# *One Stone Three Birds*: THREE-DIMENSIONAL IMPLICIT NEURAL NETWORK FOR COMPRESSION AND CONTINUOUS REPRESENTATION OF MULTI-ALTITUDE CLIMATE DATA

## ABSTRACT

Wind energy stands out as a promising clean and renewable energy alternative, not only for its potential to combat global warming but also for its capacity to meet the ever-growing demand for energy. However, analysis of wind data to fully harness the benefits of wind energy demands tackling several related challenges: (1) Current data resolution is inadequate for capturing the detailed information needed across diverse climatic conditions; (2) Efficient management and storage of real-time measurements are currently lacking; (3) Extrapolating wind data across spatial specifications enables analysis at costly-to-measure, unobserved points is necessary. In response to these challenges, we introduce a modality-agnostic learning framework utilizing implicit neural networks. Our model effectively compresses a large volume of climate data into a manageable latent codec. It also learns underlying continuous climate patterns, enabling reconstruction at any scale and supporting modality transfer and fusion. Extensive experimental results show consistent performance improvements over existing baselines.

## 1 INTRODUCTION

The escalating challenge of climate change necessitates immediate and strategic action to mitigate its impacts and steer towards sustainable development (Rolnick et al., 2022; Kaack et al., 2022). As the Earth's climate experiences higher temperatures, rising sea levels, and more extreme weather events, the transition to renewable energy sources becomes inevitable. Wind energy, in particular, offers a notable solution with its ability to deliver a clean power supply while greatly cutting down greenhouse gas emissions (Stengel et al., 2020; Ramesh et al., 2022b). However, the deployment and optimization of wind energy face several challenges:

1. *Resolution*: Identifying the most suitable sites for wind turbines necessitates data with a resolution as detailed as 1 square kilometer or finer (Irrgang et al., 2021; Kashinath et al., 2021). However, most wind farm simulations do not achieve this resolution, limiting our capacity to enhance the efficiency of wind energy farms.

2. *Data Storage*: As the granularity of simulated data and the accumulation of field measurements increase (Klöwer et al., 2021; Huang & Hoefler, 2023), the resulting growth in data size requires advanced storage solutions.

3. *Generalization*: Setting up wind measurement stations in specific locations can be challenging due to the high costs of transportation and maintenance. This necessitates cross-altitude inference, such as estimating high-altitude wind speeds from ground-level measurements.

Advancements in deep learning present promising solutions to these challenges. Techniques such as deep learning-based super-resolution can enhance low-resolution data, providing the detailed representations needed for precise analysis (Vandal et al., 2017; Gao et al., 2022; Diaconu et al., 2022). Additionally, deep learning-based data reduction can compress extensive datasets into latent formats, reducing memory and hardware demands. However, most existing deep learning approaches, like convolutional neural networks and autoencoders, are grid-based and fail to offer a continuous representation of wind fields (Nguyen et al., 2023; Requena-Mesa et al., 2021). Due to the inherently continuous nature of wind fields, there is a crucial need for methodologies that can generate

and work with continuous data representations (Reichstein et al., 2019; Luo et al., 2023; 2024b). Additionally, recent advancements in multi-modal deep learning necessitate its use in the efficient and comprehensive analysis of wind data, including the examination of wind patterns at multiple altitudes (Summaira et al., 2021; Xu et al., 2023).

In this paper, we present a novel deep learning model designed for the efficient dimension reduction and continuous reconstruction of multi-altitude climate data. Our approach utilizes modality-agnostic implicit neural networks within an *encoder-decoder (transfer)-decoder (continuous reconstructor)* framework to process multi-altitude climate data. The *encoder* segment of the three-dimensional implicit neural network functions as a nonlinear data compressor, and the *decoder (transfer)* segment functions as a modality transfer network that exploits inter-altitude data relationships to improve feature extraction. The *decoder (continuous reconstructor)* employs implicit neural representations to reconstruct continuous fields. Overall our contributions are as follows:

- We design a novel, parameter-efficient deep *encoder-decoder (transfer)-decoder (continuous reconstructor)* framework for simultaneous dimension reduction and continuous reconstruction via super-resolution of multi-altitude climate data.

- We design a three-dimensional implicit neural network for transforming data from one modality/altitude to another modality/altitude. Due to this structure of our designed implicit neural network, our proposed model is scalable to datasets consisting of a large number of modalities, unlike traditional multi-modal deep learning models which face severe scalability issues.

- We employ Gaussian Adaptive Attention Mechanism (Ioannides et al., 2024) in the *decoder (transfer)* segment of our proposed three-dimensional implicit neural network as a parameter-efficient alternative to regular query-key-value based attention mechanisms (Vaswani et al., 2017), which to the best of our knowledge has not been tested in super-resolution task.

- We employ a Kolmogorov-Arnold Network (Liu et al., 2024) as a superior alternative to traditional multi-layer perceptron neural network (Hornik et al., 1989) within the state-of-the-art Local Implicit Image Function (LIIF) (Chen et al., 2021) based *decoder (continuous reconstructor)*, which also has not been employed in data reduction or super-resolution tasks.

## 2 RELATED WORK

**Climate Downscaling** is a critical process in climate science, allowing for the translation of global climate model outputs into finer, local-scale projections. The main methods for downscaling can be categorized into dynamical downscaling and statistical downscaling (Keller et al., 2022; Harder et al., 2023). Dynamical downscaling, while comprehensive, demands significant computational power and depends on the accuracy of the global climate model data (Chau et al., 2021; Chen et al., 2022). Statistical downscaling, in contrast, is more computationally efficient and faster but assumes historical relationships will persist, potentially missing changes in climate variability and extremes (Groenke et al., 2020; Liu et al., 2020).

Alternatively, **Deep Learning-based Super-resolution** techniques are revolutionizing the enhancement of climate data, delivering unparalleled detail and precision in climate models and remote sensing imagery. Different deep neural networks specialize in analyzing both spatial and temporal climate data's complex patterns, enhancing model accuracy and details beyond what traditional downscaling achieves (Vandal et al., 2017; Requena-Mesa et al., 2021; Stengel et al., 2020; Gao et al., 2022; Diaconu et al., 2022). However, these methods often rely on fixed resolutions, highlighting the need for models that offer resolution-independent, continuous climate pattern representations (Luo et al., 2023).

**Implicit Neural Representation (INR)** uses neural networks to model continuous signals, transcending traditional discrete methods like pixel and voxel grids (Xie et al., 2022; Huang & Hoefler, 2023). This approach has recently made significant strides in climate data analysis, enabling high-resolution reconstructions beyond fixed enhancement scales. In Luo et al. (2024b), a context-aware indexing mechanism was introduced to enhance the efficiency of INR in reconstructing fields from sparse observations. In Schwarz et al. (2023), a novel compression algorithm is introduced, utilizing INR within a universal approach to data handling, that effectively generates compact yet comprehensive latent depictions of ERA5 climate data. These advancements highlight INR's capability in precise, scalable data representation. In the meantime, Neural Operator and INR have been applied

for solving PDEs and enhancing spatio-temporal resolution (Li et al., 2021; Kovachki et al., 2021). INR-based PDE solvers like MeshfreeFlowNet (Esmaeilzadeh et al., 2020) and MAgnet (Boussif et al., 2022) reconstruct continuous spatio-temporal data from sparse or discrete low-resolution inputs but lack focus on cross-spatial or cross-temporal reconstruction.

**Deep Multi-modal Learning** transforms machine perception by concurrently integrating diverse data sources such as text, images, audio, and video, unlike of traditional single-modal approaches. Multi-modal deep learning has shown impressive performance across different domains (Ngiam et al., 2011; Jing et al., 2021; Ramesh et al., 2022a; Boussioux et al., 2022; Tu et al., 2022; Cao & Gao, 2022; Ruan et al., 2023). In Qayyum et al. (2024), a novel multi-modal deep learning method is introduced for simultaneous dimension reduction and continuous cross-altitude reconstruction. However, as the number of modalities increases, the aforementioned multi-modal deep learning models encounter scalability issues, highlighting the need for a parameter-efficient multi-modal model capable of facilitating cross-altitude inference.

# 3 PRELIMINARIES

## 3.1 MULTI-ALTITUDE AS MULTI-MOADALITY

Traditionally, multi-modal data refers to the scenario of different data types from different modalities, e.g. image/text, video/text. Similar to the definition used by Qayyum et al. (2024), we follow a more flexible definition of multi-modality where data from each modality is acquired through different sensors, and we consider multi-altitude climate data as multi-modal data with each altitude as a separate modality.

## 3.2 GAUSSIAN ADAPTIVE ATTENTION MECHANISM

Conventional attention mechanism calculates weights based on the dot-product between different weight matrices (Vaswani et al., 2017), whereas Gaussian adaptive attention mechanism employs a Gaussian-based modulation of input features, enabling improvement of the standard self-attention mechanism in Transformers along with reduced number of trainable parameters and lower computational cost (Ioannides et al., 2024).

In GAAM (Gaussian Adaptive Attention Mechanism), multi-channel spatial feature $F = \{\mathbf{f}_1, \mathbf{f}_2, \cdots, \mathbf{f}_c\}$ ($c$ as the number of channels of the multi-channel spatial feature) goes through the process of computation of mean and variance:

$$\text{Mean, } \hat{\mu} = \frac{1}{c} \sum_{s=1}^{C} \mathbf{f}_s \quad (1) \qquad \text{Variance, } \hat{\sigma}^2 = \frac{1}{c} \sum_{s=1}^{c} \mathbf{f}_s^2 \quad (2)$$

The channel mean $\hat{\mu}$ is then adjusted by a learnable offset $\delta$ to learn $\psi = \delta + \hat{\mu}$. Then, the attention is computed through normalised $\mathbf{f}_s$, $\mathbf{f}_{norm,s}$:

$$\mathbf{f}_{norm,s} = \frac{\mathbf{f}_s - \psi}{\sqrt{\hat{\sigma}^2} + \epsilon} \quad (3) \qquad g_a(\mathbf{f}_s) = \exp\left(-\frac{\mathbf{f}_{norm,s}}{2\zeta^2}\right) \quad (4)$$

with a small $\epsilon > 0$. The output of the Gaussian adaptive attention block:

$$\mathbf{G}(F) = \{\mathbf{f}_1 \odot g_a(\mathbf{f}_1), \mathbf{f}_2 \odot g_a(\mathbf{f}_2), \cdots, \mathbf{f}_s \odot g_a(\mathbf{f}_s), \cdots, \mathbf{f}_c \odot g_a(\mathbf{f}_c)\} \quad (5)$$

## 3.3 THREE-DIMENSIONAL POSITIONAL ENCODER

Three-dimensional positional encoder $\mathbf{P}$ employs Fourier based positional encoding on 3D coordinate points (Tancik et al., 2020). Considering $\mathbf{v} = (x, y, h)$ to denote a 3D coordinate point, with $x^{(c)} = (x, y)$ being a point in the 2D coordinate space $\mathbf{X}_c$, and $h$ the altitude of the corresponding modality, the output of the 3D positional encoder is: $\mathbf{P}(\mathbf{v}) = [\cdots, cos(2\pi\sigma^{j/m}\mathbf{v}), cos(2\pi\sigma^{j/m}\mathbf{v}), \cdots]$ for $j \in \{0, 1, \cdots, m-1\}$, where $m$ denotes the number of frequencies of the learned Fourier features, and $\sigma$ denotes the frequency constant.

## 3.4 KOLMOGOROV-ARNOLD NETWORK

Traditional MLPs (Multi-Layer Perceptrons) model non-linearity through fixed activation functions on nodes (Hornik et al., 1989), whereas KANs employ learnable activation functions on edges (Liu et al., 2024). There is no linear weight whatsoever in KANs, each weight parameter is substituted

with a univariate function that is parameterized as a spline function instead (d. Boor, 1978). For a traditional MLP, the output at node $j$ of layer $l + 1$, can be defined as

$$x_{l+1,j} = \sigma \left( \sum_{i=1}^{n_l} w_{l+1,i,j} x_{l,i} \right) \qquad (6)$$

with $\sigma$ as a non-linear activation function. For a KAN, the output at node $j$ of layer $l + 1$, can be defined as

$$x_{l+1,j} = \sum_{i=1}^{n_l} \phi_{l+1,i,j}(x_{l,i}), \qquad (7)$$

where the non-linear function $\phi$ can be defined as $\phi(x) = w(b(x) + \text{spline}(x))$, with $b(x) = \text{silu}(x) = x/(1 + e^{-x})$ and $\text{spline}(x)$ a linear combination of B-splines, $\sum_i c_i B_i(x)$ with trainable $c_i$'s.

### 3.5 LOCAL IMPLICIT IMAGE FUNCTION

Local implicit image function (LIIF) (Chen et al., 2021) is a deep learning based continuous representation method from discrete image or multi-channel two-dimensional feature representation. A decoding function $\mathbf{D}_{\theta,liif}$ is typically parameterized as a MLP and takes the form $s = \mathbf{D}_{\theta,liif}(z, x)$, where $z$ is the observation of image pixel values or features at 2D coordinate point $x$. Consider, our objective is to predict the output at point $x^{(c)}$, with observed features $z_t$ at neighboring coordinate points $x^{(t)}, t = \{00, 01, 10, 11\}$. $S_t$ is the area of the rectangle between $x^{(c)}$ and $x^{(t')}$ where $t'$ is diagonal to $t$ (i.e. 00 to 11, 10 to 01). Then, LIIF predicts the output at $x^{(c)}$ as:

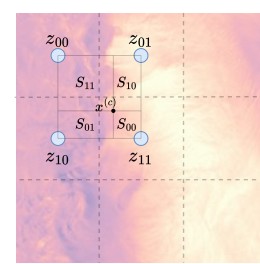

Figure 1: LIIF method.

$$s(x^{(c)}) = \sum_{t=\{00,01,10,11\}} \frac{S_t}{S} \cdot \mathbf{D}_{\theta,liif}(z_t, x^{(c)} - x^{(t)}) \qquad (8)$$

$S = \sum_t S_t$ is the total area of the four rectangles. Figure 1 illustrates this area-based interpolation through $\mathbf{D}_{\theta,liif}$. Instead of applying the area-based interpolation through $\mathbf{D}_{\theta,liif}$ directly on the input image pixels, multi-channel features $z_t$ are first extracted using an EDSR (Lim et al., 2017) model and then decoded through $\mathbf{D}_{\theta,liif}$.

## 4 PROBLEM STATEMENT

Let $\mathbf{x}_k^H \in \mathcal{M}_k^H \subset \mathbb{R}^{h \times w}$ denote the discrete high resolution data representation of modality $k$ with $\mathcal{M}_k^H$ being the discrete high resolution data space of modality $k$. We aim to achieve simultaneous data dimension reduction and cross-modal/altitude continuous reconstruction of this data instance. Our goal is to design a model capable of pertaining the following tasks:

1. **Data Dimension Reduction**: Execute $\mathbf{x}_k^H \rightarrow \mathbf{x}_k^L \in \mathcal{M}_k^L \subset \mathbb{R}^{\frac{h}{d} \times \frac{w}{d}}$, where $\mathbf{x}_k^L$ represents the discrete low resolution representation of $\mathbf{x}_k^H$ with $\mathcal{M}_k^L$ being the discrete low resolution data space of modality $k$ and $d$ being the dimension reduction factor.

2. **Cross-Altitude/Modality Continuous Reconstruction**: Execute $\mathbf{x}_k^L \rightarrow \mathbf{x}_{l \neq k}^C(x^{(c)}) \in \mathcal{M}_l^C(x^{(c)}) \subset \mathbb{R}$. Let, $x^{(c)} \in \mathbf{X}_c$ be any coordinate point in the continuous 2D coordinate space $\mathbf{X}_c$. Here $\mathbf{x}_l^C(x^{(c)})$ represents the target value at coordinate point $x^{(c)}$ for modality $l$ and $\mathcal{M}_l^C(x^{(c)})$ represents the data space of target values. The continuous nature of $\mathbf{X}_c$ makes this task a continuous reconstruction from low dimensional representation $\mathbf{x}_k^L$.

We will use cross/multi-altitude and cross/multi-modality interchangably throughout the rest of the paper.

## 5 METHOD

**Overview.** Our proposed *encoder-decoder (transfer)-decoder (continuous reconstructor)* framework, $\mathbf{T}_\theta$, comprises three primary components:

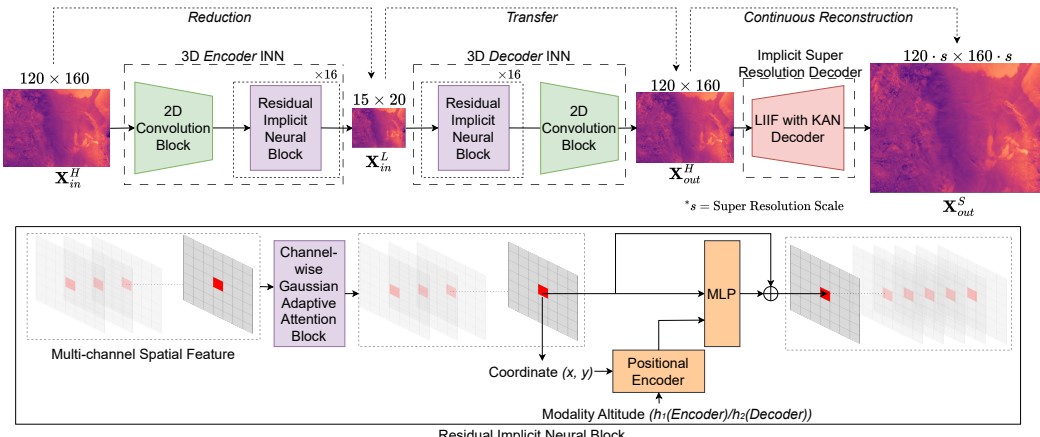

Figure 2: Overview of the proposed method, which jointly enables (a) data reduction, (b) transfer across modalities, and (c) continuous representation and arbitrary-scale super resolution.

- **encoder:** The three-dimensional implicit neural network, $\mathbf{E}_\phi$, reduces dimension of the data from input modality/altitude $\mathbf{x}_{in}^H$ to $\mathbf{x}_{in}^L$. This compression is optimized to retain sufficient information for downstream tasks. $h_{in}$ being the altitude of the input modality, we can define the encoder mapping: $\mathbf{x}_{in}^L = \mathbf{E}_\phi(\mathbf{x}_{in}^H, h_{in}), \mathbf{E}_\phi : \mathcal{M}_{in}^H \to \mathcal{M}_{in}^L$.

- **decoder (transfer):** The three-dimensional implicit neural network, $\mathbf{D}_{tr}$, transforms the discrete low resolution representation of the input modality/altitude $\mathbf{x}_{in}^L$ to the discrete high resolution representation of the target modality/altitude $\mathbf{x}_{out}^H$. $h_{out}$ being the altitude of the target modality, we can define the mapping: $\mathbf{x}_{out}^H = \mathbf{D}_{tr}(\mathbf{x}_{in}^L, h_{out}), \mathbf{D}_{tr} : \mathcal{M}_{in}^L \to \mathcal{M}_{out}^H$.

- **decoder (continuous reconstructor):** The implicit continuous decoder, $\mathbf{D}_{cr}$, that utilizes the modality/altitude-transferred discrete high resolution representation $\mathbf{x}_{out}^H$ to predict wind data at specific coordinate $\mathbf{x}_{out}^C(x^{(c)})$. We can define this mapping: $\mathbf{x}_{out}^C(x^{(c)}) = \mathbf{D}_{cr}(\mathbf{x}_{out}^H, x^{(c)}), \mathbf{D}_{cr} : \mathcal{M}_{out}^H \times \mathbf{X}_c \to \mathcal{M}_{out}^C$.

Figure 2 summarizes the proposed model, $\mathbf{T}_\theta := \mathbf{D}_{cr} \circ \mathbf{D}_{tr} \circ \mathbf{E}_\phi$. We describe the *encoder-decoder (transfer)* segment, $\mathbf{D}_{tr} \circ \mathbf{E}_\phi$ in Section 5.1 and *decoder (continuous reconstructor)* segment $\mathbf{D}_{cr}$ in Section 5.2.

## 5.1 THREE-DIMENSIONAL ENCODER-TRANSFER IMPLICIT NEURAL NETWORK

3D ETINN (Three-Dimensional Encoder-Transfer Implicit Neural Network) consists of the first two segments of the overall framework: (i) *encoder:* a 3D *encoder* INN, $\mathbf{E}_\phi$, and (ii) *decoder (transfer)*: a 3D *decoder (transfer)* INN, $\mathbf{D}_{tr}$. Both the *encoder* and *decoder (transfer)* consist of (i) a convolution block, $\mathbf{Conv}$, and several (ii) residual implicit neural blocks, $\mathbf{R}_{inn}$. 3D *encoder* INN, $\mathbf{E}_\phi$, consists of a convolutional block, $\mathbf{Conv}_{in}$, followed by several residual implicit neural blocks, $\mathbf{R}_{inn,j=\{1,2,\cdots,16\}}^{in}$, and can be described as

$$\mathbf{E}_\phi := \mathbf{R}_{inn,16}^{in} \circ \cdots \circ \mathbf{R}_{inn,2}^{in} \circ \mathbf{R}_{inn,1}^{in} \circ \mathbf{Conv}_{in} \tag{9}$$

3D *decoder (transfer)* INN, $\mathbf{D}_{tr}$, consists of several residual implicit neural blocks, $\mathbf{R}_{inn,j=\{1,2,\cdots,16\}}^{out}$, followed by a convolutional block, $\mathbf{Conv}_{out}$, and can be described as

$$\mathbf{D}_{tr} := \mathbf{Conv}_{out} \circ \mathbf{R}_{inn,16}^{out} \circ \cdots \circ \mathbf{R}_{inn,2}^{out} \circ \mathbf{R}_{inn,1}^{out} \tag{10}$$

### 5.1.1 RESIDUAL IMPLICIT NEURAL BLOCK

Residual implicit neural block $\mathbf{R}_{inn}$ consists of a Gaussian adaptive attention block $\mathbf{G}$, a three-dimensional positional encoder $\mathbf{P}$, and a 2-layer MLP network $\mathbf{D}_{inn}$. Conventional implicit neural networks use a two-dimensional positional encoder to process grid-like 2D data, such as images. In contrast, the residual implicit neural block operates on two-dimensional data while utilizing a positional encoder, $\mathbf{P}$, that accepts three-dimensional coordinate points as input. This approach allows the flexibility in transforming 2D data across modalities.

Let the input to $\mathbf{R}_{inn}$ be the multi-channel two-dimensional spatial feature, $F \in \mathbb{R}^{c \times \frac{h}{d} \times \frac{w}{d}}$, and $\mathbf{G}(F) \in \mathbb{R}^{c \times \frac{h}{d} \times \frac{w}{d}}$ be the output of the Gaussian adaptive attention block, $\mathbf{G}$. $\mathbf{G}(F)$ can be considered as a stack of $\frac{hw}{d^2}$ numbers of $c$-dimensional features, $[gf_{x^{(c)}}]_{\frac{hw}{d^2}}$ (visually illustrated in the lower block of Figure 2). Let $gf_{x^{(c)}} \in \mathbb{R}^c$ be the local feature at 2D coordinate point $x^{(c)}$ of $\mathbf{G}(F)$, where $c$ is the number of channels for the spatial feature. $gf_{x^{(c)}}$ is then transformed into $\mathbf{D}_{inn}(\{gf_{x^{(c)}}, \mathbf{P}(x, y, h)\}) + gf_{x^{(c)}}$, where $\mathbf{D}_{inn}$ is a 2-layer MLP network, $h$ is the altitude for the corresponding modality. Similar transformation is done for all those $\frac{hw}{d^2}$ numbers of $gf$'s. So, the feature transformation through $\mathbf{R}_{inn}$ can be described as a two-step transformation:

1. Feature transformation through Gaussian Adaptive Attention, $F \rightarrow \mathbf{G}(F)$.

2. Transformation of $\mathbf{G}(F)$ through residual neural network:

$$\mathbf{G}(F) = [gf_{x^{(c)}}]_{\frac{hw}{d^2}} \rightarrow \mathbf{R}_{inn}(F) = [\mathbf{D}_{inn}(\{gf_{x^{(c)}}, \mathbf{P}(x, y, h)\}) + gf_{x^{(c)}}]_{\frac{hw}{d^2}} \qquad (11)$$

## 5.2 IMPLICIT SUPER-RESOLUTION DECODER

Implicit super-resolution decoder, $\mathbf{D}_{cr} : \mathcal{M}_k^H \times \mathbf{X}_c \rightarrow \mathcal{M}_k^C$, takes the transformed discrete high-resolution representation $\mathbf{X}_k^H$ and any 2D coordinate point $x^{(c)} \in \mathbf{X}_c$ and predicts the output at coordinate point $x^{(c)}$, $\mathbf{x}_k^C(x^{(c)})$. We use a modified version of local implicit image function (LIIF) based decoder, which is a coordinate based decoding approach (Chen et al., 2021). The original LIIF decoder primarily consists of two components: (i) EDSR-based feature encoder (Lim et al., 2017), and a (ii) MLP network. In our modified LIIF-KAN decoder, we replace the MLP network with a KAN (Kolmogorov-Arnold Network)-based decoder (Liu et al., 2024).

***Decoder (continuous reconstructor)***: EDSR-based (Lim et al., 2017) feature encoder, $\mathbf{FE}$, encodes $\mathcal{M}_{out}^H$ into the encoded feature space $\mathcal{M}_{out}^F$, $\mathbf{FE} : \mathcal{M}_{out}^H \rightarrow \mathcal{M}_{out}^F$. The KAN-based decoder, $\mathbf{D}_{kan}$, then predicts the output at coordinate point $x^{(c)}$ following the similar methodology followed in LIIF (Chen et al., 2021). The overall continuous reconstruction can be described as: $\mathbf{x}_{out}^C(x^{(c)}) = (\mathbf{D}_{kan} \circ \mathbf{FE})(\mathbf{x}_{out}^H, x^{(c)})$

## 5.3 CROSS-ALTITUDE PREDICTION

Let, $\mathbf{x}_{in}^H$ be a discrete high dimensional datapoint at altitude $h_{in}$. For simultaneous dimension reduction and continuous cross-altitude reconstruction through super-resolution, we need to infer discrete low-resolution representation $\mathbf{x}_{in}^L$, and $\mathbf{x}_{out}^C(x^{(c)})$, where $x^{(c)}$ can be any 2D coordinate point in $\mathbf{X}_c$ at altitude $h_{out}$. Then,

$$\mathbf{x}_{in}^L = \mathbf{E}_\phi(\mathbf{x}_{in}^H, h_{in}) \qquad (12)$$

$$\mathbf{x}_{out}^C(x^{(c)}) = \mathbf{D}_{kan}(\mathbf{FE}(\mathbf{D}_{tr}(\mathbf{x}_{in}^L, h_{out})), x^{(c)}) \qquad (13)$$

$h_{in} \neq h_{out}$ refers to the cross-altitude prediction scenario. At a super-resolution scale $s$, we evaluate $\mathbf{D}_{KAN}$ on $s^2 hw$ number of different $x^{(c)} \in \mathbf{X}_c$, resulting in a super-resolved output with dimension $\mathbb{R}^{sh \times sw}$.

# 6 EXPERIMENTS

We first introduce the wind dataset that we used to evaluate our proposed model. Then we continue to elaborate the optimization procedure and obtained results.

## 6.1 WIND DATA

National Renewable Energy Laboratory's Wind Integration National Database (WIND) Toolkit provides high spatial and temporal resolution wind power, wind power forecasting, and meteorological data for over 126,000 locations across the continental United States during a 7-year span (Draxl et al., 2015). The simulated forecasts were developed using the Weather Research and Forecasting Model, which operates on a 2-kilometer (km) by 2-kilometer (km) grid with a 10-meter($m$) resolution from the ground to $200m$ above ground with several temporal resolutions available at $1-$hour, $4-$hour, $6-$hour, and day-ahead forecast horizons. The spatial resolution of the WIND Toolkit is 2km $\times$ 1hr in spatio-temporal resolution. As a result, the wind dataset size is 1602 (latitude) $\times$ 2976 (longitude) $\times$ 61368 (number of instances), or almost 1.2 TB per wind component (wind data at different heights). We randomly cropped data to reduce the resolution to

1500 (latitude) $\times$ 2000 (longitude) for each time instance. Wind velocity components at specific direction were determined using the wind speed and direction at a specific height. For example, if wind speed at height $h$ is $V$ at an angle $\theta^\circ$ with northern direction, the northern and eastern components are $u = V\cos\theta^\circ, v = V\sin\theta^\circ$ accordingly. In this paper, we only report the results with the northern projection of wind speed data.

## 6.2 EXPERIMENTAL SETUP

**Dataset:** To evaluate the continuous reconstruction capability of our proposed method, we created a dataset for multi-modal super-resolution tasks using simulated wind data. From 61368 instances, we randomly sampled 1500 data points from various timestamps at heights of $10m, 60m, 160m$ and $200m$. We used 1200 data points for training and 300 for testing. We used bicubic interpolation to generate a pair of discrete and continuous high-resolution samples for each instance. For example, if the discrete input dimension is $(120 \times 160)$ and the super-resolution scale is $2.5\times$, the continuous output dimension is $(300 \times 400)$. During training, the continuous high-resolution sample is created by cropping from the actual $(1500 \times 2000)$ resolution data and then generating the discrete high-resolution sample via bicubic interpolation. The discrete high-resolution dimension was set to $(120 \times 160)$, with super-resolution scales ranging from $1\times$ to $3\times$. To avoid randomness during testing, both continuous and discrete high-resolution samples were generated by bicubic interpolation from the actual $(1500 \times 2000)$ data.

**Training Details:** Adam was adopted as the optimizer (Kingma & Ba, 2017), to train the model for 600 epochs. The learning rate during the optimization was set following the cyclical learning rate technique with a minimum learning rate of $10^{-5}$ and a maximum learning rate of $10^{-4}$ (Smith, 2017). At super-resolution scale $s$, the super-resolution ground truth $\mathbf{x}_{out}^S$ has a dimension of $sh \times sw$. Coordinate-based implicit neural networks, e.g. LIIF (Chen et al., 2021), predict outputs pixel-by-pixel instead of predicting the whole super-resolved image at a single forward pass. We randomly selected 2048 coordinate points, at each optimization step, among the $s^2hw$ coordinate points of the target super-resolved continuous representation, and optimized the parameters based on the predictions $\hat{\mathbf{x}}_{out}^C(x^{(c)})$ and the ground truth $\mathbf{x}_{out}^C(x^{(c)})$ on those coordinate points to extradite the optimization process. We used the $L_1$ loss function, $l_1(\hat{\mathbf{x}}_{out}^C, \mathbf{x}_{out}^C, x^{(c)}) = |\mathbf{x}_{out}^C(x^{(c)}) - \hat{\mathbf{x}}_{out}^C(x^{(c)})|$, to optimize the model. It took about 12 hours to train the model on a workstation with a single NVIDIA A100 GPU.

**Evaluation Metrics:** We employed two metrics to evaluate the continuous reconstruction through super resolution. Peak signal-to-noise ratio (PSNR) is the ratio of a signal's maximum possible value (power) to the power of distorting noise that affects the quality of its representation. Structural similarity index (SSIM) is a perceptual metric that evaluates the degradation of image quality, that compares the spatial structures between the target image and reproduced image. For evaluation of the compression performance, we employed Compression Ratio (CR) metric which measures the ratio between the required memory for storing the compressed versus the uncompressed data. These evaluation metrics have been discussed in detail in the Appendix Section C.

## 6.3 EXPERIMENT 1: SUPER-RESOLUTION

**Task.** We evaluated the model's continuous reconstruction capability by performing super-resolution at different scales on a test dataset containing 300 data points.

**Setup.** The high-resolution input dimension was set to $120 \times 160$, with a dimension reduction factor $d = 8$, yielding a low-resolution representation of $15 \times 20$. We focus on cross-altitude predictions where the input modality height $h_{in}$ is closer to the

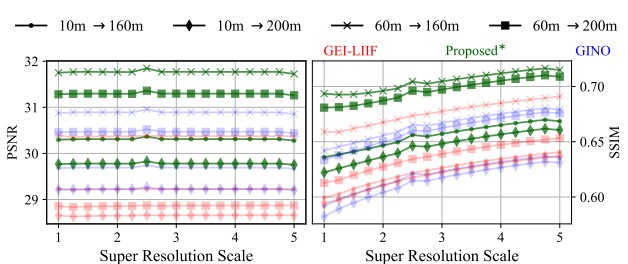

Figure 3: Results(Experiment 1)

ground (10m, 60m) and the output modality height $h_{out}$ is significantly higher (160m, 200m). Due to the lack of existing methodologies for simultaneous dimension reduction and reconstruction through super-resolution, we use the following two methods as baselines:

- **Multi-Altitude GEI-LIIF:** Multi-altitude simultaneous reduction and continuous reconstruction through GEI-LIIF method proposed by Qayyum et al. (2024).

- **GINO:** Bicubic downscaling followed by Geometry-Informed Neural Operator (GINO) (Li et al., 2024).

**Result Summary.** As shown in Figure 3, our model consistently outperforms both the baselines across all cross-altitude scenarios and super-resolution scales. Performance is evaluated using two metrics, and results are plotted with varying super-resolution scales on the x-axis, demonstrating superior performance across the board. Results for super-resolution scales $s \in [1, 3]$ and $s \in (3, 5]$ respectively illustrate the results for in-distribution scales and out-of-distribution scales.

## 6.4 EXPERIMENT 2: DATA COMPRESSION

**Task.** We evaluate the data compression performance of our model and compare it against existing compression methods, focusing on cross-altitude predictions and reconstruction accuracy.

**Setup.** We employed the Prediction by Partial Matching (PPM) data compression algorithm with $\mu$-law encoding at various quantization levels ($Q$) for data compression and reconstruction, as described in Moffat (1990). Additionally, we tested bicubic interpolation for compressing and decompressing the data. For cross-altitude predictions, we used the wind power law $v_1/v_2 = (h_1/h_2)^\alpha$ to transform the reconstructed data from one height to another for both PPM and bicubic methods. To assess performance, we measured the average compression ratio, PSNR, and SSIM over the test set. Since traditional compression methods lack super-resolution capabilities, we limit super-resolution comparison to the scale $s = 1$ for our model.

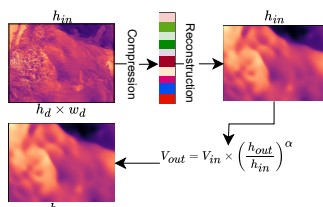

Figure 4: Data compression followed by cross-altitude prediction using wind power law.

**Result Summary.** Figure 4 illustrates the cross-altitude prediction methodology of these methods, while Table 1 presents a detailed comparison. For cross-altitude predictions where the input modality height is 10m, our model outperforms the baseline methods across all three metrics—compression ratio, PSNR, and SSIM. In cases where the input modality height is 60m, while our model does not achieve the best PSNR and SSIM, it significantly surpasses others in terms of compression ratio. For bicubic models with a reduction factor $d = 4$, the low-resolution dimension is $30 \times 40$. With a higher reduction factor ($d = 8$), bicubic models achieve a compression ratio similar to ours but at the cost of lower PSNR and SSIM. Likewise, PPM models attain higher compression ratios at lower $Q$ values, but at the expense of PSNR and SSIM. Only our model consistently delivers a high compression ratio alongside either the best or near-best PSNR and SSIM. Detailed comparison of data compression performances is provided in Table 3 of the Appendix Section G.

Table 1: Comparative compression performance at different cross-altitude prediction scenarios.

| | $\mathcal{H}_{in} = 10m \rightarrow \mathcal{H}_{out} = 160m$ | | | $\mathcal{H}_{in} = 10m \rightarrow \mathcal{H}_{out} = 200m$ | | |
|---|---|---|---|---|---|---|
| Method | PSNR ↑ | SSIM ↑ | CR ↑ | PSNR ↑ | SSIM ↑ | CR ↑ |
| $\text{PPM}_{Q=16,\alpha=0.16}$ | 28.3741 | 0.6147 | 92.3746 | 27.5924 | 0.5755 | 92.3746 |
| $\text{Bicubic}_{d=4,\alpha=0.16}$ | 29.4141 | 0.6157 | 93.7500 | 28.7553 | 0.5914 | 93.7500 |
| Proposed Method* | 30.2967 | 0.6360 | 98.4375 | 29.7689 | 0.6222 | 98.4375 |
| | $\mathcal{H}_{in} = 60m \rightarrow \mathcal{H}_{out} = 160m$ | | | $\mathcal{H}_{in} = 60m \rightarrow \mathcal{H}_{out} = 200m$ | | |
| Method | PSNR ↑ | SSIM ↑ | CR ↑ | PSNR ↑ | SSIM ↑ | CR ↑ |
| $\text{PPM}_{Q=16,\alpha=0.16}$ | 30.0943 | 0.7205 | 93.2051 | 29.2434 | 0.6793 | 93.2051 |
| $\text{Bicubic}_{d=4,\alpha=0.16}$ | 31.7323 | 0.7168 | 93.7500 | 31.0304 | 0.6917 | 93.7500 |
| Proposed Method* | 31.7500 | 0.6934 | 98.4375 | 31.2823 | 0.6808 | 98.4375 |

## 6.5 EXPERIMENT 3: MODALITY TRANSFER

**Task.** We evaluate the performance of our model without the 3D ETINN segment by designing and training separate models for different altitude levels, then comparing them in cross-altitude prediction tasks.

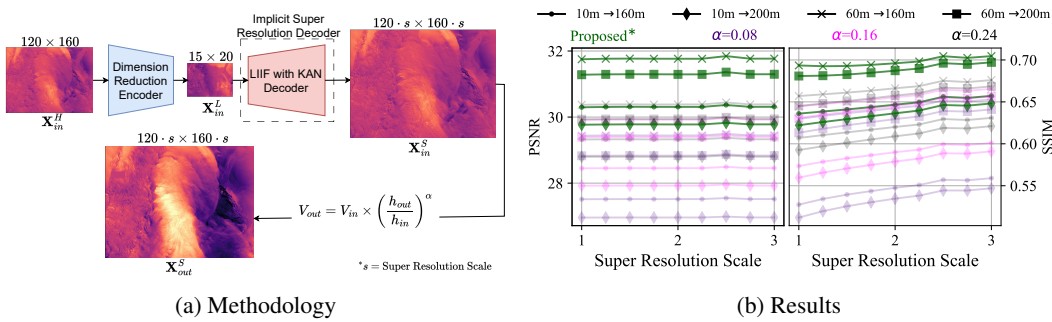

(a) Methodology                    (b) Results

Figure 5: (a) Methodology and (b) Results for Experiment 3.

**Setup.** We removed the 3D ETINN segment and developed four distinct models, each trained on data from a specific height, following consistent optimization procedures. For cross-altitude predictions, we used the wind power law (Touma, 1977) to transform the reconstructed data from one altitude to another. For instance, for an input height $h_{in} = 10m$ and output height $h_{out} = 200m$, we applied the model trained at 10m to predict super-resolved wind data and then used the wind power law to transform it to 200m. Figure 5a summarizes this cross-altitude prediction methodology without the 3D ETINN. The downsampling architecture for dimension reduction followed the design of the invertible UNet (Etmann et al., 2020), as depicted in Figure 5a. We tested different values of $\alpha$ for cross-altitude predictions at various super-resolution scales.

**Result Summary.** Figure 5b presents the average PSNR and SSIM across the test set at different super-resolution scales. In all cross-altitude prediction scenarios, the model incorporating the 3D ETINN consistently outperforms the models without it, demonstrating the critical role of the 3D ETINN in enhancing performance.

## 6.6 EXPERIMENT 4: ABLATION STUDIES

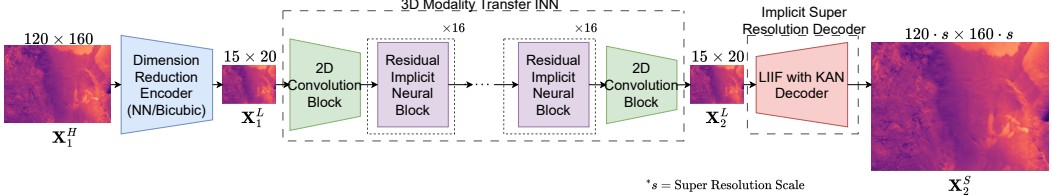

Figure 6: The integrated NN-based and bicubic interpolation-based dimension reduction encoders.

**Task 1: Ablation of Decoder.** We evaluate the performance of the proposed decoder by comparing it against other implicit neural network-based super-resolution models.

**Setup.** To benchmark the performance of our model, we replaced the LIIF-KAN decoder segment with the LIIF (Chen et al., 2021), ITNSR (Yang et al., 2021), HiNOTE (Luo et al., 2024a), SRNO (Wei & Zhang, 2023), DIINN (Nguyen & Beksi, 2023) and LTE (Lee et al., 2022) models. We also tried LTE-KAN, replacing MLP with KAN in the LTE model, similar to the modfication done in LIIF-KAN. All models were trained using the same optimization procedures. Figure 7 illustrates the average PSNR and SSIM

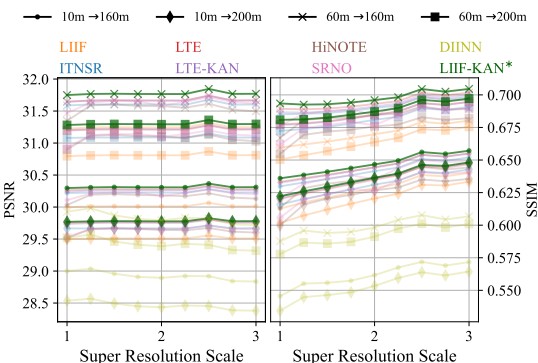

Figure 7: Ablation of decoders.

across the test set at varying super-resolution scales, with the x-axes of each plot showing the respective scales.

**Result Summary.** Our modified LIIF-based decoder, LIIF-KAN, consistently outperforms all other implicit super-resolution models, including the original LIIF decoder. The results show a significant performance improvement in both PSNR and SSIM, demonstrating that LIIF-KAN offers superior super-resolution capabilities compared to existing models.

**Task 2: Ablation of Encoder.** We evaluate the effectiveness of our 3D encoder INN by comparing it with other dimension reduction encoders.

**Setup.** To benchmark performance, we replaced our 3D *encoder* INN with two alternative encoders: a Neural Network (NN)-based encoder and a bicubic interpolation-based encoder. The design of the NN-based encoder follows the downsampling architecture from the invertible UNet model (Etmann et al., 2020), as outlined in Figure 6. We trained all encoders using the same procedures, and Figure 8 presents the average PSNR and SSIM results across the test set at various super-resolution scales, shown on the x-axis of each plot.

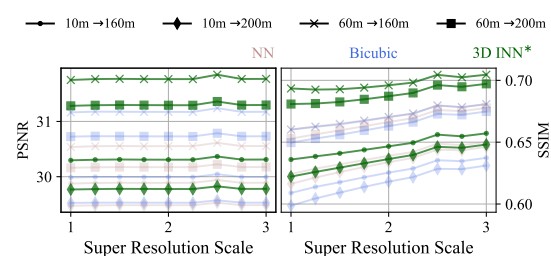

Figure 8: Ablation of encoders.

**Result Summary.** The results demonstrate the comparative performance of each dimension reduction encoder. Our 3D *encoder* INN consistently achieves higher PSNR and SSIM scores than the NN-based and bicubic interpolation-based encoders, particularly at higher super-resolution scales, affirming its superior performance for dimension reduction tasks.

**Task 3: Ablation of Attention Mechanisms.** We assess the efficacy of the Gaussian adaptive attention in the Residual Implicit Neural blocks by comparing it with other attention mechanisms.

**Setup.** We replaced the Gaussian adaptive attention mechanism in the Residual Implicit Neural blocks with the standard Query, Key, Value (QKV) attention mechanism (Vaswani et al., 2017), and also tested the performance without any attention mechanism. All models were trained following identical procedures. Figure 9 presents the results, showing the performance of each variant.

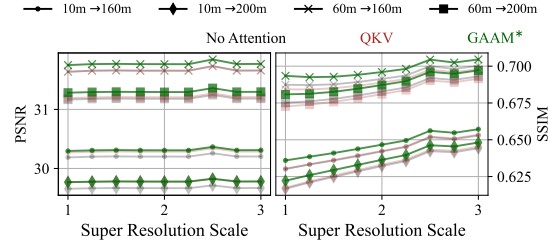

Figure 9: Ablation of attention mechanism.

**Result Summary.** The results clearly demonstrate that our proposed method, incorporating the Gaussian adaptive attention mechanism, consistently outperforms both the QKV-based attention and the version without attention. This highlights the superior effectiveness of the Gaussian adaptive attention in enhancing model performance.

## 7 CONCLUSION

We have developed an innovative deep learning approach for simultaneous continuous super-resolution, data dimensionality reduction, and multi-altitude learning for climatological data. We designed a three-dimensional implicit neural network specifically for learning continuous, rather than discrete, representations of multi-altitude velocity fields used for wind farm power modeling across the continental United States. Unlike traditional multi-modal deep learning models, which handle only a limited number of modalities due to scalability issues, our three-dimensional implicit neural network is scalable to large number of modalities. We employed two very recently proposed techniques: Gaussian adaptive attention mechanism and Kolmogorov-Arnold network to design our model and modify existing models. Experimental results have shown promising potential in improving wind energy assessment for electricity generation, efficient large data storage via dimensionality reduction, and extrapolation to inaccessible spatial areas. Both Gaussian adaptive attention and Kolmogorov-Arnold networks have the potential to enhance interpretability in machine learning models, making this an area for future research.

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

## A HYPERPARAMETER DETAILS

Table 2: List of hyperparamters.

| Three-Dimensional Implicit Neural Network | | | | |
|---|---|---|---|---|
| Residual Implicit Neural Block | | | 16 | |
| Residual Implicit Neural Block | | | | |
| GAAM | | Positional Encoder | MLP | |
| Feature | 64 | $\sigma$ | 25.0 | Hidden Dimension | 256 |
| Head | 1 | $m$ | 30 | Layer | 2 |
| Gaussian Function | 1 | | | Output Dimension | 64 |

| Implicit Super Resolution Decoder | | | |
|---|---|---|---|
| EDSR Feature Encoder | | KAN Decoder | |
| Feature | 64 | Hidden Layer | 2 |
| Residual Block | 16 | Neuron | 32 |
| Residual Scale | 1 | Grid size | 5 |
| | | Spline Order | 3 |
| | | Scale Noise | 0.1 |
| | | Base Scale | 1.0 |
| | | Spline Scale | 1.0 |
| | | Grid Epsilon | 0.02 |
| | | Grid Range | $[-1, 1]$ |

## B LIST OF NOTATIONS

| | |
|---|---|
| $\mathbf{x}_k^H$ | Discrete high resolution data representation of modality $k$ |
| $\mathcal{M}_k^H$ | Discrete high resolution data space of modality $k$ |
| $\mathbf{x}_k^L$ | Discrete low resolution data representation of modality $k$ |
| $\mathcal{M}_k^L$ | Discrete low resolution data space of modality $k$ |
| $\mathbf{X}_c$ | Two-dimensional coordinate space |
| $\mathbf{x}^{(c)}$ | Two-dimensional coordinate point |
| $\mathcal{M}_l^C(x^{(c)})$ | Data space of target value at coordinate point $x^{(c)}$ for modality $l$ |

## C EVALUATION METRICS

**Peak Signal to Noise Ratio(PSNR):**   PSNR measures the ratio between the peak signal power and noise power present in the signal. For a super resolved predicted output $\hat{\mathbf{x}}$, and its corresponding gournd-truth $\mathbf{x}$, the peak signal to noise ratio is calculated using the following formula:

$$\mathbf{PSNR}(\mathbf{x}, \hat{\mathbf{x}}) = 10 \log_{10} \left( \frac{(\mathrm{MAX}(\mathbf{x}) - \mathrm{MIN}(\mathbf{x}))^2}{\frac{1}{s^2 hw} \sum_{x^c} (\mathbf{x}(x^{(c)}) - \hat{\mathbf{x}}(x^{(c)}))^2} \right)$$

**Structural Similarity Index Measure(SSIM):**   SSIM measures the similarity between the similarity between two images. SSIM is widely used metric for evaluation of super resolution of images or uniform 2D grid like data. For a super resolved predicted output $\hat{\mathbf{x}}$, and its corresponding gournd-truth $\mathbf{x}$, the structural similarity index measure between the window $w_x$ of $\mathbf{x}$ and window $w_{\hat{x}}$ is calculated using the following formula:

$$\mathbf{SSIM}(\mathbf{x}_{w_x}, \hat{\mathbf{x}}_{w_{\hat{x}}}) = \frac{(2\mu_x \mu_{\hat{x}} + c_1)(2\sigma_{x\hat{x}} + c_2)}{(\mu_x^2 + \mu_{\hat{x}}^2 + c_1)(\sigma_x^2 + \sigma_{\hat{x}}^2 + c_2)}$$

Here,

- $\mu_x =$ Sample mean of window $w_x$
- $\mu_{\hat{x}} =$ Sample mean of window $w_{\hat{x}}$
- $\sigma_x^2 =$ Variance of window $w_x$
- $\sigma_{\hat{x}^2} =$ Variance of window $w_{\hat{x}}$
- $\sigma_{x\hat{x}} =$ Covariance of windows $w_x$ and $w_{\hat{x}}$
- $c_1 = (k_1 L)^2, c_2 = (k_2 L)^2$
- $k_1 = 0.01, k_2 = 0.03$
- $L$ is the range of values for $\mathbf{x}$ and $\hat{x}$, $L = 2$ is chosen as the range is $[-1, 1]$

**Compression Ratio(CR):** Compression ratio is the widely used metric for evaluation of data compression. For the compressed data representation $\mathbf{x}_{comp}$, and its corresponding uncompressed data representation $\mathbf{x}$, the compression ration can be calculated using the following formula:

$$\mathbf{CR} = \left(1 - \frac{\text{memory size of } \mathbf{x}_{comp}}{\text{memory size of } \mathbf{x}}\right) \times 100$$

## D  EXPERIMENT: SPATIAL GENERALIZATION

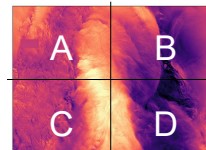

Figure 10: Regions

**Task: Region-based Generalization.** We investigate the spatial generalization performance of our model across different regions of the data for super-resolution tasks.

**Setup.** The original dataset has a spatial dimension of $1500 \times 2000$, which we partitioned into four distinct regions: A, B, C, and D, each sized $750 \times 1000$ (illustrated in Figure 10). Four separate models were trained on data from each of these regions, and we evaluated their super-resolution performance across all regions, regardless of the model's training source. Figure 11 depicts the super-resolution outcomes for each model tested on each region.

**Result Summary.** The results reveal a clear regional bias: models trained on a specific region perform best when tested on data from that same region. For instance, the model trained on region A exhibits superior performance when evaluated on test data from region A, as shown in Figure 11. Similar trends are observed for the other regions. These findings underscore the need for future work to focus on improving spatial generalization to enhance the ability of models to simultaneously perform data dimensionality reduction and cross-modal reconstruction through super-resolution techniques.

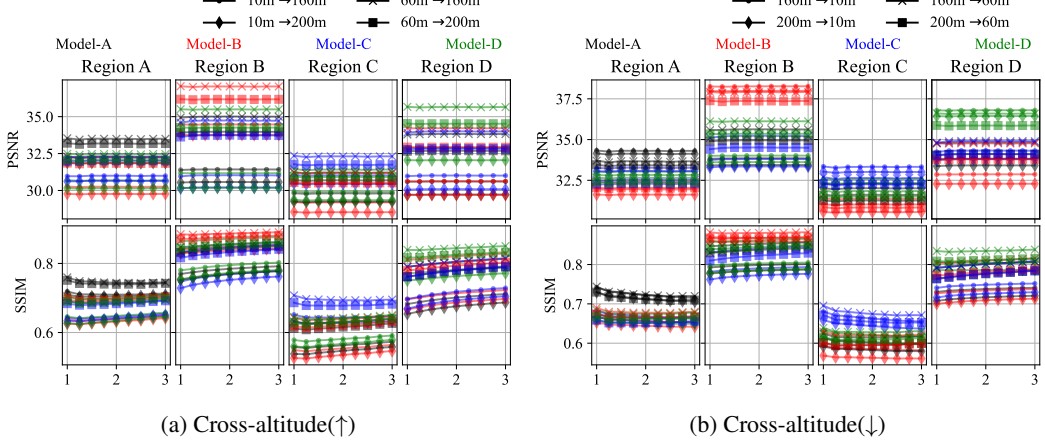

(a) Cross-altitude($\uparrow$)    (b) Cross-altitude($\downarrow$)

Figure 11: Results for spatial generalization test.

**Task: Altitude-based Generalization.** We investigate the altitude-based generalization performance of our model for super-resolution tasks.

**Setup.** The original dataset used in this paper has data from four different altitudes $h = \{10m, 60m, 160m, 200m\}$. However, we want evaluate our model's performance in zero-shot learning for cross-altitude continuous reconstruction. To evaluate this generalization test, we trained four different models with single-altitude excluded dataset. For example, one model is trained with data from altitudes $h = \{10m, 60m, 160m\}$, that is the model does not see any data from altitude $h = 200m$. Then this model's performance is evaluated in continuous cross-altitude reconstruction on data from altitude $h = 200m$. Similarly, same experiments are conducted on other altitudes.

**Result Summary.** Our proposed approach exhibits potential for further enhancement in altitude-based zero-shot learning. Models demonstrate superior performance at altitudes included in the training set compared to those where data from the target altitude was absent during training. The results of altitude-based generalization tests are illustrated in Figure 12.

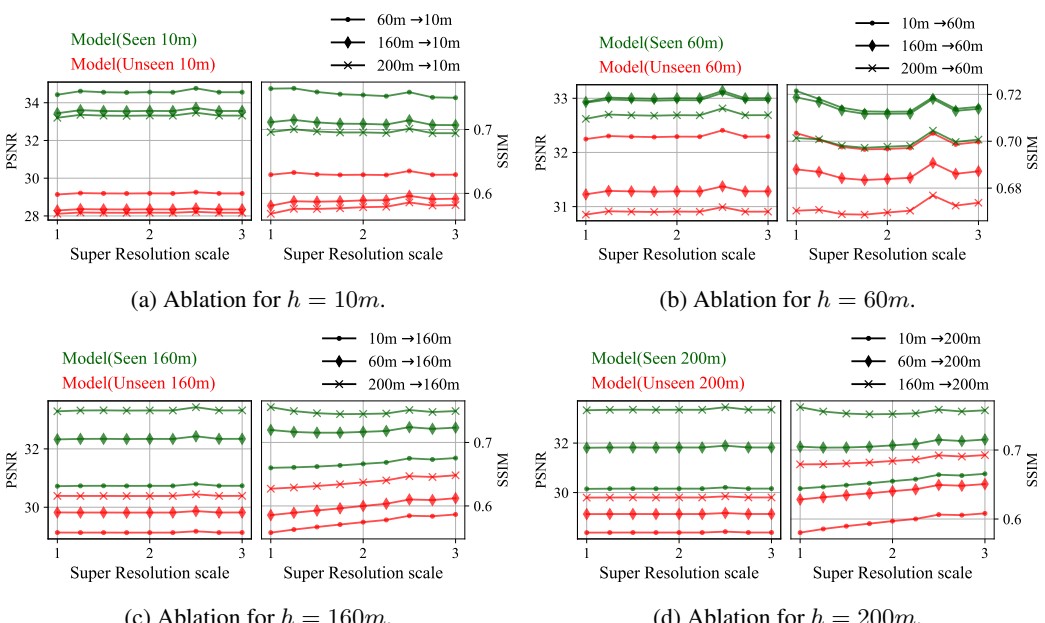

Figure 12: Results for altitude generalization test.

# E ADDITIONAL COMPARATIVE RESULTS

**Baseline Experimental Setup:** The experimental setup for the two baseline methods used in this paper is discussed below:

- **GEI-LIIF:** Unlike our proposed approach, the baseline GEI-LIIF method is not scalable to higher number of data modalities. Even though the GEI-LIIF method can be designed to work with 4 modalities, optimizing such a GEI-LIIF model is an infeasible approach. Instead, we trained 4 different models, each trained with 2 modalities, where the pairs are formed with data from altitudes: $(10m, 160m)$, $(10m, 200m)$, $(60m, 160m)$, and $(60m, 200m)$.

- **GINO:** For the Geometry-Informed Neural Operator (GINO) baseline, we use bicubic downsampling for downscaling the discrete high-resolution input data to discrete low-resolution representations and then employ a GINO model (Li et al., 2024) for continuous reconstruction from the low-resolution representations. The GINO model takes the low-resolution representations and their corresponding three-dimensional coordinate points (altitude as the third axis) as its input and predicts the output at target three-dimensional coordinate points.

**Results:** Figure 13a shows that our proposed method outperforms the baseline GEI-LIIF method in those cross-altitude prediction scenarios where the height of the input modality is higher above from the ground and the height of the output modality is closer to ground. Likewise, Figure 13b shows that our proposed method outperforms the baseline GEI-LIIF method in intra-altitude prediction scenarios too. Different super-resolution scales are shown at the $x-$axes of each plot.

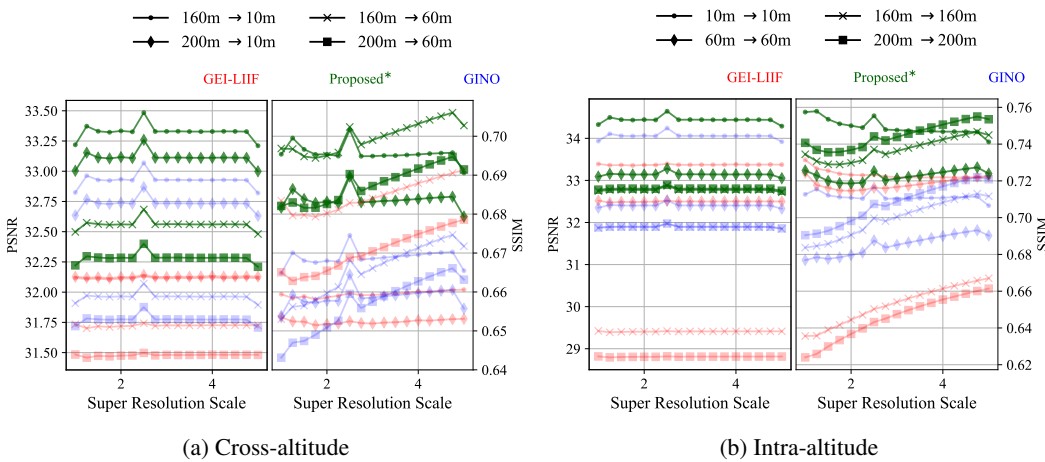

(a) Cross-altitude

(b) Intra-altitude

Figure 13: Comparative super resolution performance of our proposed method with baseline.

# F ADDITIONAL ABLATION EXPERIMENTS

We also did some additional experiments with the Gaussian Adaptive Attention Mechanism within the 3D *encoder-transfer* INN segment of our proposed approach. Results for this additional experiment, also additional results of the previous ablation experiments is discussed here.

**Implicit Super Resolution Decoder:** We show the super resolution performance of different super resolution decoders in the cross-altitude prediction scenarios where the height of the input modality is much higher above from the ground and the height of the output modality is closer to the ground. Figure 14a shows our proposed LIIF-KAN super resolution decoder outperforms other implicit neural network based super resolution decoders in these scenarios also. Likewise, Figure 14b shows our proposed LIIF-KAN super resolution decoder outperforms its counterparts in intra-altitude prediction scenarios too.

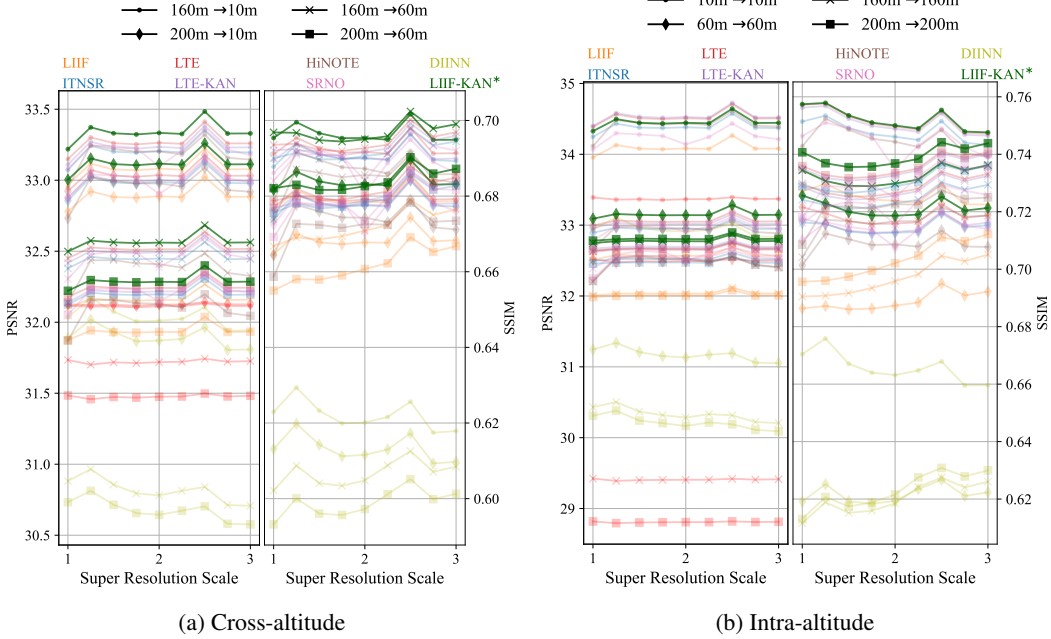

(a) Cross-altitude

(b) Intra-altitude

Figure 14: Comparative super resolution performance with ablation experiments of *Decoder (continuous reconstructor)*.

**Modality Transfer:** We removed the 3D ETINN segment of our proposed model and designed 4 different models for 4 different heights, and trained these models following the same optimization procedure of our model. For comparison of cross-altitude predictions, we used the wind power law to transform the reconstructed data at one height to another height. Figure 15a summarizes the average PSNR and SSIM over the test set for different super resolution scales in those cross-altitude scenarios where the height of the input modality is much higher above from the ground and the height of the output modality is closer to the ground. Unlike the inverse cross modal scenarios, the models without 3D MTINN performs better than our proposed model in most of these cross modal prediction scenarios. Figure 15b shows similar superior performances in intra-altitude prediction scenarios.

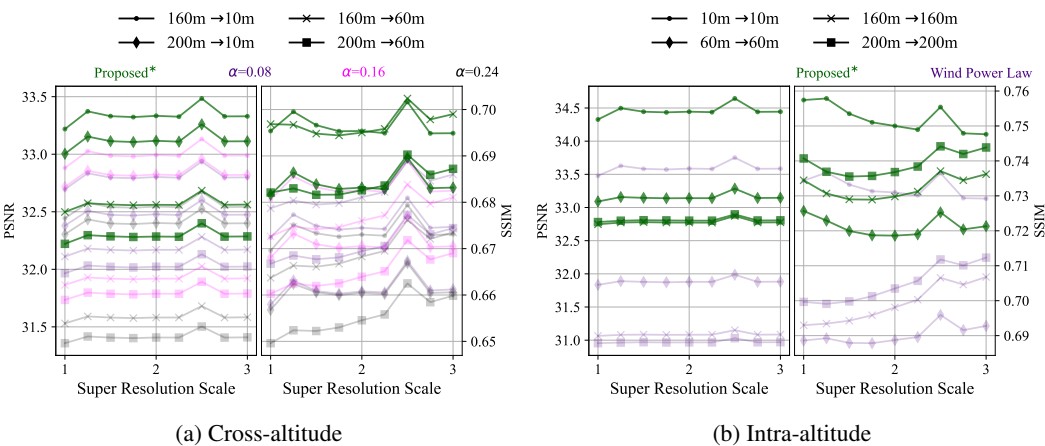

(a) Cross-altitude                    (b) Intra-altitude

Figure 15: Comparative super resolution performance with ablation experiments of 3D ETINN.

**Dimension Reduction Encoder:** Figure 16a summarizes the average PSNR and SSIM over the test set for different dimension reduction encoders at different super resolution scales for those cross-altitude prediction scenarios where the height of input modality is higher above from the ground and the height of output modality is closer to the ground. Similar results are observed in intra-altitude prediction scenarios as shown in Figure 16b.

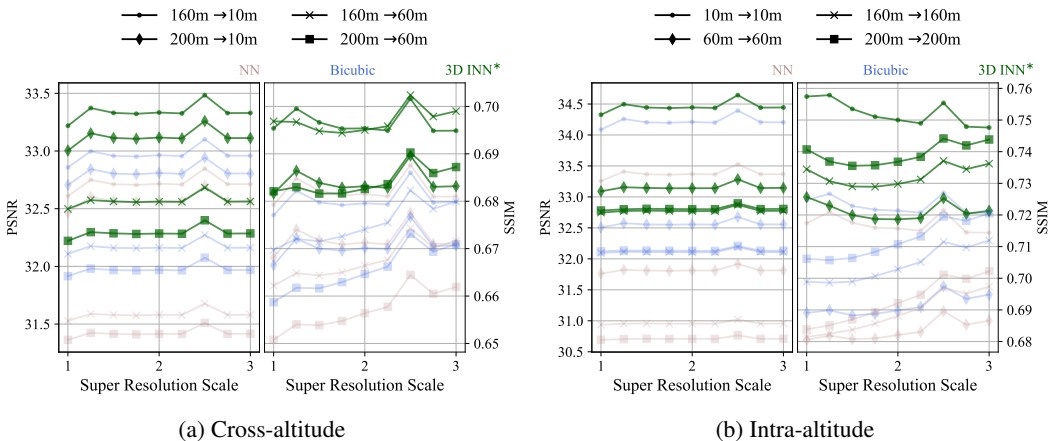

(a) Cross-altitude                    (b) Intra-altitude

Figure 16: Comparative super resolution performance with ablation experiments of dimension reduction encoders.

**Attention Mechanism:** Figure 17 summarizes the average PSNR and SSIM over the test set for different attention mechanisms in 3D INN at different super resolution scales for those cross-altitude prediction scenarios where the height of input modality is higher above from the ground and the height of output modality is closer to the ground, and also in intra-altitude prediction scenarios.

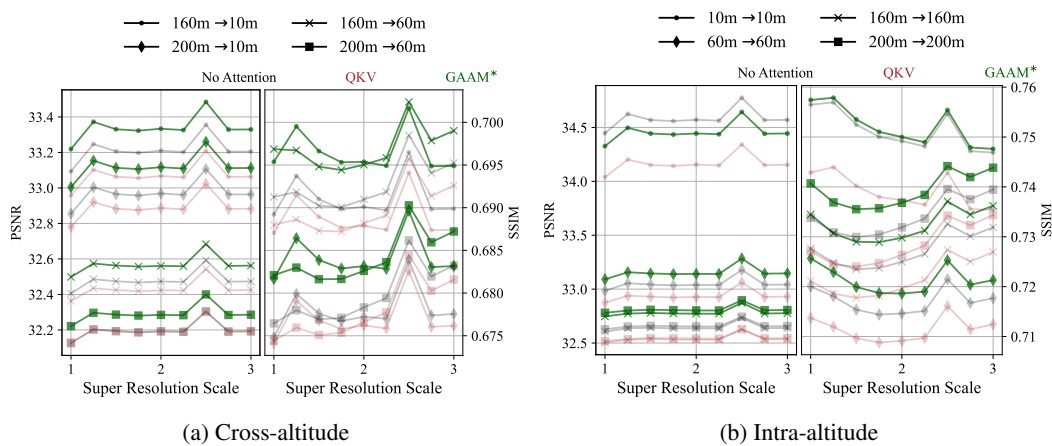

(a) Cross-altitude

(b) Intra-altitude

Figure 17: Comparative super resolution performance with ablation experiments of attention mechanisms.

**Gaussian Adaptive Attention Mechanism:** We adjusted the number of Gaussian functions in the Gaussian Adaptive Attention blocks of the 3D *encoder-transfer* INN within our proposed model architecture. All the results in this work as our proposed method uses single Gaussian function. Figure 18 shows the super resolution performances of our proposed method with different number of Gaussian functions. The results indicate minimal performance variation across models with differing numbers of Gaussian functions. In some cross-altitude predictions, the model with 8 Gaussian functions underperforms slightly compared to others. However, aside from this minor difference, the performance across models remains largely consistent. Notably, increasing the number of Gaussian functions results in longer inference times. Consequently, the variant with a single Gaussian function was chosen to minimize computational cost.

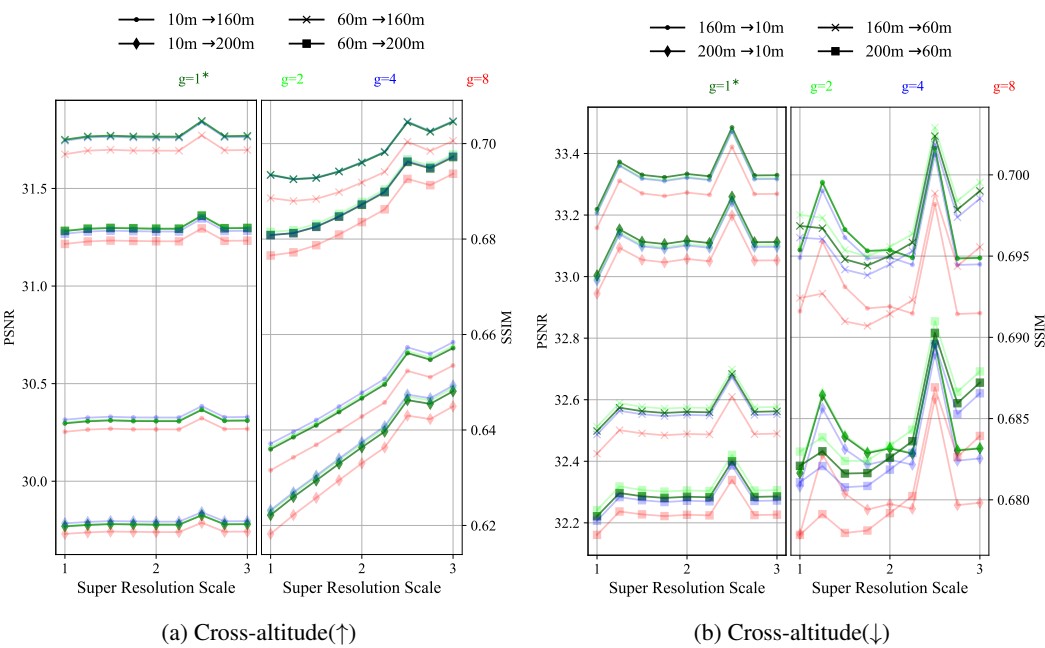

(a) Cross-altitude(↑)

(b) Cross-altitude(↓)

Figure 18: Comparative super resolution performance with ablation experiments of attention mechanisms.

## G  DATA COMPRESSION PERFORMANCE

We tested our approach and compared its compression performance with other data compression methods. We used prediction by the partial matching (PPM) data compression algorithm with the $\mu$-law based encoding at different quantization levels ($Q$) to compress and reconstruct data. We also tested bicubic interpolation to compress and decompress the data. For comparison of cross-modal predictions, we used the wind power law to transform the reconstructed data at one height to another height with the PPM and bicubic methods. Table 3 shows comparative performance of these compression models and our proposed model.

| Method | $\mathcal{H}_{in} = 10m \to \mathcal{H}_{out} = 160m$ | | | $\mathcal{H}_{in} = 10m \to \mathcal{H}_{out} = 200m$ | | |
|---|---|---|---|---|---|---|
| | PSNR ↑ | SSIM ↑ | CR ↑ | PSNR ↑ | SSIM ↑ | CR ↑ |
| $\text{PPM}_{Q=4,\alpha=0.08}$ | 14.7861 | 0.1834 | 96.5459 | 14.7896 | 0.1801 | 96.5459 |
| $\text{PPM}_{Q=8,\alpha=0.08}$ | 25.7107 | 0.4375 | 94.9431 | 25.2910 | 0.4165 | 94.9431 |
| $\text{PPM}_{Q=12,\alpha=0.08}$ | 27.8809 | 0.5647 | 93.3153 | 27.2358 | 0.5331 | 93.3153 |
| $\text{PPM}_{Q=16,\alpha=0.08}$ | 29.1510 | 0.6313 | 92.3746 | 28.3580 | 0.5952 | 92.3746 |
| $\text{PPM}_{Q=4,\alpha=0.16}$ | 14.7959 | 0.1908 | 96.5459 | 14.8000 | 0.1879 | 96.5459 |
| $\text{PPM}_{Q=8,\alpha=0.16}$ | 24.1419 | 0.4104 | 94.9431 | 23.6673 | 0.3893 | 94.9431 |
| $\text{PPM}_{Q=12,\alpha=0.16}$ | 26.6262 | 0.5402 | 93.3153 | 25.9743 | 0.5067 | 93.3153 |
| $\text{PPM}_{Q=16,\alpha=0.16}$ | 28.3741 | 0.6147 | 92.3746 | 27.5924 | 0.5755 | 92.3746 |
| $\text{PPM}_{Q=4,\alpha=0.24}$ | 14.7908 | 0.1948 | 96.5459 | 14.7914 | 0.1915 | 96.5459 |
| $\text{PPM}_{Q=8,\alpha=0.24}$ | 22.1256 | 0.3789 | 94.9431 | 21.5227 | 0.3568 | 94.9431 |
| $\text{PPM}_{Q=12,\alpha=0.24}$ | 24.6256 | 0.4971 | 93.3153 | 23.8819 | 0.4603 | 93.3153 |
| $\text{PPM}_{Q=16,\alpha=0.24}$ | 26.5045 | 0.5704 | 92.3746 | 25.6755 | 0.5264 | 92.3746 |
| $\text{Bicubic}_{d=8,\alpha=0.08}$ | 28.5727 | 0.5453 | 98.4375 | 27.9979 | 0.5292 | 98.4375 |
| $\text{Bicubic}_{d=4,\alpha=0.08}$ | 29.1551 | 0.6073 | 93.7500 | 28.4866 | 0.5845 | 93.7500 |
| $\text{Bicubic}_{d=8,\alpha=0.16}$ | 28.6436 | 0.5506 | 98.4375 | 28.0937 | 0.5341 | 98.4375 |
| $\text{Bicubic}_{d=4,\alpha=0.16}$ | 29.4141 | 0.6157 | 93.7500 | 28.7553 | 0.5914 | 93.7500 |
| $\text{Bicubic}_{d=8,\alpha=0.24}$ | 27.6741 | 0.5341 | 98.4375 | 27.0468 | 0.5144 | 98.4375 |
| $\text{Bicubic}_{d=4,\alpha=0.24}$ | 28.4513 | 0.5961 | 93.7500 | 27.7140 | 0.5961 | 93.7500 |
| Proposed Method[*] | 30.2967 | 0.6360 | 98.4375 | 29.7689 | 0.6222 | 98.4375 |

| Method | $\mathcal{H}_{in} = 60m \to \mathcal{H}_{out} = 160m$ | | | $\mathcal{H}_{in} = 60m \to \mathcal{H}_{out} = 200m$ | | |
|---|---|---|---|---|---|---|
| | PSNR ↑ | SSIM ↑ | CR ↑ | PSNR ↑ | SSIM ↑ | CR ↑ |
| $\text{PPM}_{Q=4,\alpha=0.08}$ | 13.0141 | 0.2383 | 96.9004 | 13.0370 | 0.2354 | 96.9004 |
| $\text{PPM}_{Q=8,\alpha=0.08}$ | 25.1141 | 0.4977 | 95.5889 | 24.8110 | 0.4782 | 95.5889 |
| $\text{PPM}_{Q=12,\alpha=0.08}$ | 28.2098 | 0.6374 | 94.0468 | 27.6855 | 0.6050 | 94.0468 |
| $\text{PPM}_{Q=16,\alpha=0.08}$ | 30.6537 | 0.7296 | 93.2051 | 29.8449 | 0.6907 | 93.2051 |
| $\text{PPM}_{Q=4,\alpha=0.16}$ | 13.0192 | 0.2454 | 96.9004 | 13.0436 | 0.2438 | 96.9004 |
| $\text{PPM}_{Q=8,\alpha=0.16}$ | 24.6593 | 0.4904 | 95.5889 | 24.2708 | 0.4699 | 95.5889 |
| $\text{PPM}_{Q=12,\alpha=0.16}$ | 27.5075 | 0.6259 | 94.0468 | 26.8810 | 0.5913 | 94.0468 |
| $\text{PPM}_{Q=16,\alpha=0.16}$ | 30.0943 | 0.7205 | 93.2051 | 29.2434 | 0.6793 | 93.2051 |
| $\text{PPM}_{Q=4,\alpha=0.24}$ | 13.0240 | 0.2523 | 96.9004 | 13.0498 | 0.2517 | 96.9004 |
| $\text{PPM}_{Q=8,\alpha=0.24}$ | 24.1300 | 0.4824 | 95.5889 | 23.6195 | 0.4605 | 95.5889 |
| $\text{PPM}_{Q=12,\alpha=0.24}$ | 26.7056 | 0.6118 | 94.0468 | 25.9287 | 0.5739 | 94.0468 |
| $\text{PPM}_{Q=16,\alpha=0.24}$ | 29.3359 | 0.7071 | 93.2051 | 28.3628 | 0.6619 | 93.2051 |
| $\text{Bicubic}_{d=8,\alpha=0.08}$ | 30.3704 | 0.6305 | 98.4375 | 29.8115 | 0.6151 | 98.4375 |
| $\text{Bicubic}_{d=4,\alpha=0.08}$ | 31.6032 | 0.7161 | 93.7500 | 30.8552 | 0.6915 | 93.7500 |
| $\text{Bicubic}_{d=8,\alpha=0.16}$ | 30.3703 | 0.6299 | 98.4375 | 29.8462 | 0.6144 | 98.4375 |
| $\text{Bicubic}_{d=4,\alpha=0.16}$ | 31.7323 | 0.7168 | 93.7500 | 31.0304 | 0.6917 | 93.7500 |
| $\text{Bicubic}_{d=8,\alpha=0.24}$ | 30.1536 | 0.6257 | 98.4375 | 29.5740 | 0.6084 | 98.4375 |
| $\text{Bicubic}_{d=4,\alpha=0.24}$ | 31.5744 | 0.7128 | 93.7500 | 30.8145 | 0.6853 | 93.7500 |
| Proposed Method[*] | 31.7500 | 0.6934 | 98.4375 | 31.2823 | 0.6808 | 98.4375 |

Table 3: Comprehensive comparative compression performance.

