# OpenReview forum: "$\textit{One Stone Three Birds:}$ Three-Dimensional Implicit Neural Network for Compression and Continuous Representation of Multi-Altitude Climate Data"
_ICLR.cc/2025/Conference — Submitted to ICLR 2025_

### Official Review · Reviewer_HTHi · 2024-10-28

**Soundness:** 2
**Presentation:** 2
**Contribution:** 2
**Rating:** 3
**Confidence:** 5

**Summary:**

This paper designed a deep learning model to achieve scalable multi-modality learning, data dimensionality reduction, and continuous super-resolution for climate data. It addresses three emergent issues related to wind power generation: insufficient resolution of wind data, lack of storage of wind data, and high cost of wind data gathering at specific locations.

**Strengths:**

originality: It provides a new architecture for continuous super-resolution, which makes multi-modality learning more scalable.

quality: It provides rather comprehensive experiments to support its architecture modification and superiority.

clarity: The paper is readable to catch its main idea, but to get deep understanding of technique details requires effort.

significance: It addresses wind power generation challenges claimed in the paper.

**Weaknesses:**

Not enough justification of its architecture modification even though experimentally supported.

The notation system is extremely hard to follow.

The paper requires extra effort on the writing.

Your model is capable of arbitrary scale super-resolution after training. The model should be tested for zero-shot super-resolution which means to test on a different upsampling factor from the one it was trained on.

Your model is a super-resolution model. But there is not extensive comparison to other super-resolution models.

**Questions:**

Line 60: the framework name is very confusing.
Line 134: It is hard to read and understand. It requires to rephrase.
Figure 1: The figure is too busy and hard to digest. For example, there are many dashed lines intersecting
L289: You did not split a part of your data into validation set. Then how did you choose your model hyper parameters?
L289: Why would you choose a rather small dataset for your method? Is it enough to train your model?
L296: why are training data and test data constructed differently?

---

> ### Author Response · Authors · 2024-11-20
> **Response to Initial Comments and Concerns Raised**
>
> We thank reviewer HTHi for the comments. We would like to respond to some of the issues the reviewer mentioned:
>
> _Justification for architectural modification:_ We have updated the manuscript with justifications for the architectural choices:
>  - _3D INN encoder-decoder:_ This makes our model scalable to multi-altitude data, unlike the baseline method where the number of model parameters increases with the increasing number of modalities/altitudes.
>  - _GAAM:_ Traditional self-attention mechanism is a parameter and computation heavy approach, whereas Gaussian adaptive attention can achieve superior performance with fewer parameters and with lower computational cost.
>  - _KAN:_ For implicit neural networks, the last layer is traditionally a MLP. Recently, KAN has emerged as a superior alternative to MLPs in many applications, especially better capturing complex functional relationships in continuous space with better model complexity. That is the motivation behind replacing the last MLP layer of the implicit neural network decoder with KAN and empirical results show that this modification improves the performance.
>
> _Notation update:_ Thank you for the suggestion. We have revised the problem statement, methodology section, added a preliminaries section, and also added a list of notations in the Appendix Section B to make the notations easy to follow.
>
> _Out of distribution scales:_ Updated Figure 3 with out-of-distribution super-resolution scales.
>
> _Zero-shot learning/Spatial generalization:_ Added new experimental results on altitude-based spatial generalizability test. Due to space constraints, we have moved the section on spatial generalizability section to Appendix Section D.
>
> _Extensive comparison to other super-resolution models:_ We have compared our proposed LIIF-KAN with more super-resolution models, including:
>  - HiNOTE ([1] ICML 2024)
>  - SRNO ([2] CVPR 2023)
>  - DIINN ([3] WCAV 2023)
>  - LTE-KAN ([4] CVPR 2022)
>
> along with the previous baselines. As our problem statement specifically requires continuous reconstruction, we only focus on existing continuous super-resolution models as baselines.
>
> [1] Luo, X., Qian, X. and Yoon, B.J., Hierarchical Neural Operator Transformer with Learnable Frequency-aware Loss Prior for Arbitrary-scale Super-resolution. In Forty-first International Conference on Machine Learning.
>
> [2] Wei, M. and Zhang, X., 2023. Super-resolution neural operator. In Proceedings of the IEEE/CVF Conference on Computer Vision and Pattern Recognition (pp. 18247-18256).
>
> [3] Nguyen, Q.H. and Beksi, W.J., 2023. Single image super-resolution via a dual interactive implicit neural network. In Proceedings of the IEEE/CVF Winter Conference on Applications of Computer Vision (pp. 4936-4945).
>
> [4] Lee, J. and Jin, K.H., 2022. Local texture estimator for implicit representation function. In Proceedings of the IEEE/CVF conference on computer vision and pattern recognition (pp. 1929-1938).
>
> _Selection of framework name:_ The framework comprises three main components: a 3D INN encoder for dimensionality reduction, a 3D INN decoder for modality transfer between low-resolution representations of input and output modalities (altitude), and an implicit super-resolution decoder that predicts outputs at arbitrary points. This _encoder-decoder(transfer)-decoder(continuous reconstructor)_ structure is detailed in the Methodology section, with revised phrasing incorporated in the manuscript.
>
> _Figures:_ Updated congested figures for improved visibility and readability.
>
> _Validation set:_ The test data has been treated as the validation data in this case. Hyperparameters related to KAN, LIIF were set following their original works. Other hyperparameters were selected by the performances on the test data.
>
> _Dataset size:_ The baseline method mentioned in this work used a similar amount of data. Compared to traditional image data for super-resolution, the number of samples might seem small, but each sample has a dimension of $1500\times2000$, which is enough for splitting one sample into several data samples and treat each splitted sample as separate data point (although we did not do so in the training phase, we just randomly cropped a segment from the original $1500\times2000$ sample). Due to this comparatively large size of the actual sample in the dataset, compared to the discrete high resolution sample of size $120\times160$, this dataset with a seemingly small number of samples is sufficient to optimize the parameters of a deep learning model.
>
> _Different processing of test and train data:_ At the training phase, the high resolution sample is extracted arbitrarily from the actual high resolution data with a dimension of $1500\times2000$. To avoid randomness at the testing phase, we used bicubic downsampling of the entire $1500\times2000$ data point to extract the high-resolution sample.
>
> We have updated our manuscript and submitted the revised version. Please let us know if you have any questions or suggestions.

---

> ### Author Response · Authors · 2024-11-24
> **Additional SOTA ML Baselines for Comparison**
>
> We would like to re-address one recent development with manuscript revision. According to the suggestion from one of the reviewers on additional baseline experiment, we have added Geometry-Informed Neural Operator based experimental results in the revised manuscript.
>
> _Additional SOTA ML Baseline:_ Our work tackles three tasks—data compression, modality transfer, and super-resolution—simultaneously. To our knowledge, GEI-LIIF is a comparable work with the highest similarity to our problem and is included as a baseline. We thank the reviewers for suggesting additional baseline experiment with Fourier Neural Operators. We chose Geometry-Informed Neural Operator [5] as the additional baseline experiment due to its suitability for this specific problem statement, and updated the manuscript with results of this additional baseline experiment.
>
> [5] Li, Zongyi, et al. "Geometry-informed neural operator for large-scale 3d pdes." Advances in Neural Information Processing Systems 36 (2024).
>
> Please let us know if you have any further questions, comments or suggestions.

---

> ### Author Response · Authors · 2024-11-27
> **Follow-up on Revised Paper Submission**
>
> Dear Reviewer,
>
> We hope this message finds you well. As the deadline for manuscript revision is approaching to the end, we wanted to follow up to check if you have had a chance to review the revised version of the paper. Please let us know if there are any additional revisions or clarifications required on our part. We would be happy to address them promptly.
>
> Thank you for your time and feedback!
>
> Regards,
>
> Authors

---

### Official Review · Reviewer_Ztio · 2024-11-01

**Soundness:** 3
**Presentation:** 3
**Contribution:** 3
**Rating:** 6
**Confidence:** 4

**Summary:**

Authors introduce a multi-modal architecture inspired by the implicit neural representation literature for cross-modal super-resolution. Their architecture's design is motivated by its usefulness for wind profile prediction, which tackles the following three points:
1. **Resolution**: They build a continuous representation which allows their implicit super-resolution decoder to infer the wind field at any given coordinate.
2. **Data storage**: Their 3D Encoder Implicit Neural Network (INN) allows them to downsample the input low-resolution wind field that is later used for continuous reconstruction, which provides memory savings and efficient storage.
3. **Generalization**: Their achitecture allows for modality transfer using ther 3D Decoder INN which transfers the downsampled input wind field at one altitude level to a different altitude with the same resolution as the input, allowing them to later continuously reconstruct the higher-resolution wind profile.

**Strengths:**

- The paper is well motivated and presents three challenges in the face of wind energy deployment, which guided the design of their proposed architecture.
- Extensive experiments for demonstrating the strength of the architecture for super-resolution, compression and modality transfer
- OOD experiments for cross-region wind profile prediction
- Extensive ablation experiments motivate the utility of introduced architectural components

**Weaknesses:**

- Previous works have introduced architectures for doing similar things albeit in the related PDE solving and Neural Operator literature and the authors neither cite them nor do they compare against (at least one of) them:
  - [MeshfreeFlowNet: A Physics-Constrained Deep Continuous Space-Time  Super-Resolution Framework (SC20: International Conference for High Performance Computing, Networking, Storage and Analysis 2020)](https://arxiv.org/pdf/2005.01463.pdf)
  - [MAgNet: Mesh Agnostic Neural PDE Solver (Advances in Neural Information Processing Systems 2022)](https://arxiv.org/pdf/2210.05495.pdf)
  - [Fourier Neural Operator for Parametric Partial Differential Equations (International Conference On Learning Representations 2020)](https://arxiv.org/pdf/2010.08895.pdf)
 To name a few

**Questions:**

# Questions
- In line 302, it's said that the model is optimized using the L1 loss, however, it's not clear what exactly are the inputs of this loss. It's a loss between what and what?
# Comments
- The paper doesn't contain visualizations of the upscaled frames. Regression learns the mean of the conditional distribution which leads to a loss of the high-frequency information that is usually present in the wind profile (especially with turbulence).
- It's a bit hard to parse the PSNR/SSIM figure since there's no space between them, it would be better to have a small vertical space between them.

---

> ### Author Response · Authors · 2024-11-20
>
> We thank reviewer Ztio for the comments. We would like to respond to some of the issues the reviewer mentioned:
>
> _FNO or INR based PDE solvers for tentative baselines:_ Neural Operator and INR have been applied for solving PDEs and enhancing spatio-temporal resolution. INR-based PDE solvers like MeshfreeFlowNet and MAgnet reconstruct continuous spatio-temporal data from sparse or discrete low-resolution inputs but do not focus on cross-spatial or cross-temporal reconstruction. Specifically, MeshfreeFlowNet does resemble with our proposed approach. They employ a UNet-based context encoder for upscaling the low resolution spatio-temporal data, and then employ an INR model for continuous reconstruction from the context. Our _encoder-decoder(transfer)_ segment can be considered similar to a context encoder. However, these PDE solvers do not focus on cross-spatial or cross-temporal inference. We have updated our manuscript including these papers as related work. In addition, we have added two neural operator-based models in our latest manuscript: SRNO and HiNOTE. They represent SOTA neural operator-based SR models. In the future, we also plan to modify and employ some FNO-based approaches for cross-altitude continuous reconstruction.
>
> _$L_1$ loss:_ While training, the model is evaluated on a set of sampled coordinate points (at any continuous spatial coordinates instead of fixed regular grids in other existing methods) For example, let’s say the model is asked to predict the wind speed at point $(x, y)$ and altitude $h$. Let the ground-truth wind speed at that coordinate is $v(x, y, h)$ and the model’s prediction is $\hat{v}(x, y, h)$. Then the objective is to minimize $L_1(v, \hat{v})$. Description of this has been added in the revised manuscript.
>
> _Loss of high-frequency information:_ The model is expected to lose some of the high-frequency information even when the super-resolution scale is 1. The reason behind this is the fact that we are reconstructing the data from a reduced dimensional data. So even if the super-resolution scale is 1.0, technically the super-resolution scale is 8.0 because we first downscale the data and then reconstruct the data from this reduced dimensional space. This very feature of simultaneous dimension reduction and arbitrary scale super resolution makes the problem even more challenging.
>
> _Neat figures:_ Thank you for the suggestion. We have updated all the figures in the revised manuscript with hopefully improved figures for more visibility and readability.
>
> According to the valuable comments and suggestions from all the reviewers, we have updated our manuscript and submitted the revised version. Please let us know if you have any questions or suggestions.

---

> ### Author Response · Authors · 2024-11-21
> **Updated Model with Fourier Neural Operator**
>
> We would like to thank the reviewer for suggesting experiment with Fourier Neural Operators. In the _encoder-decoder(transfer)_ segment of our proposed model, we replaced the Residual Implicit Neural Block with Fourier Neural Operator and trained the model with similar optimization procedures. Here, attached the results on the two evaluation metrices for different super resolution scales:
>
> **10m to 160m**
>
> _PSNR:_
>
> | Method    |1.0x|1.25x|1.5x |1.75x|2.0x |2.25x| 2.5x |2.75x|3.0x |
> | -------- | ------- | ------- | ------- | ------- | ------- | ------- | ------- | ------- |------- |
> | Proposed*  | 30.2967| 30.3071|30.3117| 30.3084| 30.308| 30.3078|30.366 |30.3096| 30.3108|
> | FNO  | 23.6252| 23.6255| 23.6262| 23.6258| 23.6258| 23.626 | 23.6382| 23.626 | 23.6265|
>
> _SSIM:_
>
> | Method    |1.0x|1.25x|1.5x |1.75x|2.0x |2.25x| 2.5x |2.75x|3.0x |
> | -------- | ------- | ------- | ------- | ------- | ------- | ------- | ------- | ------- |------- |
> | Proposed*  | 0.636 | 0.6385| 0.641 | 0.6438| 0.6466| 0.6495| 0.6561| 0.6547|0.6572|
> | FNO  | 0.0875| 0.0913| 0.0942| 0.0963| 0.0983| 0.1  | 0.1021| 0.1027|0.1038|
>
>
> **10m to 200m**
>
> _PSNR:_
>
> | Method    |1.0x|1.25x|1.5x |1.75x|2.0x |2.25x| 2.5x |2.75x|3.0x |
> | -------- | ------- | ------- | ------- | ------- | ------- | ------- | ------- | ------- |------- |
> | Proposed*  | 29.7689| 29.7758| 29.7802| 29.7786| 29.7772| 29.7772| 29.8255|29.7791| 29.7799|
> | FNO  | 23.2344|23.2336|23.2346|23.2342|23.2342|23.2344|23.2449|23.2346|23.2349|
>
> _SSIM:_
>
> | Method    |1.0x|1.25x|1.5x |1.75x|2.0x |2.25x| 2.5x |2.75x|3.0x |
> | -------- | ------- | ------- | ------- | ------- | ------- | ------- | ------- | ------- |------- |
> | Proposed*  | 0.6222| 0.626 | 0.6294| 0.6329| 0.6364| 0.6397| 0.6463| 0.6455|0.6481|
> | FNO  | 0.0826| 0.0864| 0.0894|0.0915|0.0934|0.0952|0.0972|0.0978|0.0989|
>
>
> **60m to 160m**
>
> _PSNR:_
>
> | Method    |1.0x|1.25x|1.5x |1.75x|2.0x |2.25x| 2.5x |2.75x|3.0x |
> | -------- | ------- | ------- | ------- | ------- | ------- | ------- | ------- | ------- |------- |
> | Proposed*  | 31.75  | 31.7667| 31.7706| 31.7667| 31.7659| 31.7656| 31.8459|31.7679| 31.7697|
> | FNO  | 23.6252| 23.6255| 23.6262| 23.6258| 23.6258| 23.626 | 23.6382| 23.626 | 23.6265|
>
> _SSIM:_
>
> | Method    |1.0x|1.25x|1.5x |1.75x|2.0x |2.25x| 2.5x |2.75x|3.0x |
> | -------- | ------- | ------- | ------- | ------- | ------- | ------- | ------- | ------- |------- |
> | Proposed*  | 0.6934| 0.6925| 0.6928| 0.6941| 0.696 | 0.6982| 0.7045| 0.7025|0.7046|
> | FNO  | 0.0875| 0.0913| 0.0942| 0.0963| 0.0983| 0.1  | 0.1021| 0.1027|0.1038|
>
>
> **60m to 200m**
>
> _PSNR:_
>
> | Method    |1.0x|1.25x|1.5x |1.75x|2.0x |2.25x| 2.5x |2.75x|3.0x |
> | -------- | ------- | ------- | ------- | ------- | ------- | ------- | ------- | ------- |------- |
> | Proposed*  | 31.2823| 31.2934| 31.2974| 31.2956| 31.2936| 31.2935| 31.3607|31.296 | 31.2973|
> | FNO  | 23.2344|23.2336|23.2346|23.2342|23.2342|23.2344|23.2449|23.2346|23.2349|
>
> _SSIM:_
>
> | Method    |1.0x|1.25x|1.5x |1.75x|2.0x |2.25x| 2.5x |2.75x|3.0x |
> | -------- | ------- | ------- | ------- | ------- | ------- | ------- | ------- | ------- |------- |
> | Proposed*  | 0.6808| 0.6812| 0.6826| 0.6847| 0.6872| 0.6898| 0.6962| 0.6948|0.6972|
> | FNO  | 0.0826| 0.0864| 0.0894|0.0915|0.0934|0.0952|0.0972|0.0978|0.0989|
>
> Please let us know if you have any questions or suggestions.

---

> > ### Comment · Reviewer_Ztio · 2024-11-22
> >
> > Thank you for incoporating my comments and suggestions, it's very much appreciated!
> >
> > > We would like to thank the reviewer for suggesting experiment with Fourier Neural Operators. In the encoder-decoder(transfer) segment of our proposed model, we replaced the Residual Implicit Neural Block with Fourier Neural Operator and trained the model with similar optimization procedures. Here, attached the results on the two evaluation metrices for different super resolution scales:
> >
> > I must admit that I am quite surprised by FNO's low performance on wind prediction. Alongside the metrics, I would appreciate it if the authors could also release visual predictions. What I had in mind is a direct comparison with FNO, rather than ablations of your architecture, which you have already demonstrated. While FNO is not INR-based and cannot predict the wind profile at arbitrary coordinates, I still believe it is valuable as a baseline, given that you predict the wind profile on a grid.
> >
> > FNO can still be used even without a time series of observations. If you prefer to compare against an INR-based approach, I suggest using MeshFreeFlowNet.
> >
> > Regardless of the baseline chosen, you could train it to predict a higher-resolution wind profile at a different altitude, without including a transfer module and using the original architecture as is, to demonstrate that your approach is indeed superior for super-resolution across different modalities.
> >
> > I am open to revising my score if the experimental results from comparisons with either baseline demonstrate that your architecture offers a novel contribution.

---

> > > ### Author Response · Authors · 2024-11-22
> > > **FNO-based Baseline**
> > >
> > > We would like to thank the reviewer for the thoughtful suggestions.
> > >
> > > _Potential FNO-based baseline:_ As a suitable _full_-baseline, we believe [Geometry-Informed Neural Operator for Large-Scale 3D PDEs](https://proceedings.neurips.cc/paper_files/paper/2023/hash/70518ea42831f02afc3a2828993935ad-Abstract-Conference.html) can be thought of as a proper baseline to evaluate our work. We are currently conducting experiment, where we are using bicubic interpolation to downscale the discrete high resolution input data, and GINO as the continuous reconstructor from the discrete low resolution data. The _geometry-informed_ nature makes this model inherently capable of transferring information from the input altitude to the target altitude. We will report the results once the experiments are done.
> > >
> > > _MeshFreeFlowNet:_ As we mentioned earlier, our proposed model is the closest to the architecture of MeshFreeFlowNet. Our _encoder-decoder(transfer)_ segment works like an alternative to the 3D UNet-based context encoder of MeshFreeFlowNet, and our LIIF-KAN based _decoder(continuous reconstructor)_ has similar functionalities as that of the continuous decoder of MeshFreeFlowNet. However, MeshFreeFlowNet is not inherently capable of cross-modal/cross-axis/cross-channel information transfer, leaving us to the choice for an alternative with cross-modal information transfer capabilities, such as our proposed 3D Implicit Neural Network.
> > >
> > > Please let us know if you have any further questions or suggestions.

---

> > > > ### Comment · Reviewer_Ztio · 2024-11-22
> > > >
> > > > Thank you for your response. Indeed, GINO is a more suitable baseline in this case. I have no further questions or suggestions at the moment and will report back once the GINO results are available.

---

> > > > > ### Author Response · Authors · 2024-11-23
> > > > > **Results for GINO**
> > > > >
> > > > > We would like to thank the reviewer once again for suggesting experiment with Geometry-Informed Neural Operator. We would like to report the results for in-distribution super resolution scales:
> > > > >
> > > > > **10m to 160m**
> > > > >
> > > > > _PSNR:_
> > > > >
> > > > > | Method    |1.0x|1.25x|1.5x |1.75x|2.0x |2.25x| 2.5x |2.75x|3.0x |
> > > > > | -------- | ------- | ------- | ------- | ------- | ------- | ------- | ------- | ------- |------- |
> > > > > | Proposed*  | 30.2967| 30.3071|30.3117| 30.3084| 30.308| 30.3078|30.366 |30.3096| 30.3108|
> > > > > | GINO  | 29.6795| 29.6904| 29.6928| 29.6911| 29.6919| 29.6908| 29.7403 | 29.6916|29.6924|
> > > > >
> > > > > _SSIM:_
> > > > >
> > > > > | Method    |1.0x|1.25x|1.5x |1.75x|2.0x |2.25x| 2.5x |2.75x|3.0x |
> > > > > | -------- | ------- | ------- | ------- | ------- | ------- | ------- | ------- | ------- |------- |
> > > > > | Proposed*  | 0.636 | 0.6385| 0.641 | 0.6438| 0.6466| 0.6495| 0.6561| 0.6547|0.6572|
> > > > > | GINO  | 0.5918| 0.5978| 0.6023| 0.6067| 0.6107| 0.6142 | 0.621 | 0.6204 | 0.6231|
> > > > >
> > > > >
> > > > > **10m to 200m**
> > > > >
> > > > > _PSNR:_
> > > > >
> > > > > | Method    |1.0x|1.25x|1.5x |1.75x|2.0x |2.25x| 2.5x |2.75x|3.0x |
> > > > > | -------- | ------- | ------- | ------- | ------- | ------- | ------- | ------- | ------- |------- |
> > > > > | Proposed*  | 29.7689| 29.7758| 29.7802| 29.7786| 29.7772| 29.7772| 29.8255|29.7791| 29.7799|
> > > > > | GINO  | 29.2148| 29.2211| 29.224 | 29.2238| 29.2237| 29.2227| 29.2644|29.2237|29.2243|
> > > > >
> > > > > _SSIM:_
> > > > >
> > > > > | Method    |1.0x|1.25x|1.5x |1.75x|2.0x |2.25x| 2.5x |2.75x|3.0x |
> > > > > | -------- | ------- | ------- | ------- | ------- | ------- | ------- | ------- | ------- |------- |
> > > > > | Proposed*  | 0.6222| 0.626 | 0.6294| 0.6329| 0.6364| 0.6397| 0.6463| 0.6455|0.6481|
> > > > > | GINO  | 0.5824| 0.5893| 0.5947| 0.5997| 0.6041| 0.608 | 0.6147| 0.6146|0.6175|
> > > > >
> > > > >
> > > > > **60m to 160m**
> > > > >
> > > > > _PSNR:_
> > > > >
> > > > > | Method    |1.0x|1.25x|1.5x |1.75x|2.0x |2.25x| 2.5x |2.75x|3.0x |
> > > > > | -------- | ------- | ------- | ------- | ------- | ------- | ------- | ------- | ------- |------- |
> > > > > | Proposed*  | 31.75  | 31.7667| 31.7706| 31.7667| 31.7659| 31.7656| 31.8459|31.7679| 31.7697|
> > > > > | GINO  | 30.8753| 30.8905| 30.892 | 30.8901| 30.8909| 30.8893|30.9541|30.8903|30.8914|
> > > > >
> > > > > _SSIM:_
> > > > >
> > > > > | Method    |1.0x|1.25x|1.5x |1.75x|2.0x |2.25x| 2.5x |2.75x|3.0x |
> > > > > | -------- | ------- | ------- | ------- | ------- | ------- | ------- | ------- | ------- |------- |
> > > > > | Proposed*  | 0.6934| 0.6925| 0.6928| 0.6941| 0.696 | 0.6982| 0.7045| 0.7025|0.7046|
> > > > > | GINO  | 0.6419| 0.6456| 0.6486| 0.6521| 0.6555| 0.6587| 0.6654| 0.6644|0.667|
> > > > >
> > > > >
> > > > > **60m to 200m**
> > > > >
> > > > > _PSNR:_
> > > > >
> > > > > | Method    |1.0x|1.25x|1.5x |1.75x|2.0x |2.25x| 2.5x |2.75x|3.0x |
> > > > > | -------- | ------- | ------- | ------- | ------- | ------- | ------- | ------- | ------- |------- |
> > > > > | Proposed*  | 31.2823| 31.2934| 31.2974| 31.2956| 31.2936| 31.2935| 31.3607|31.296 | 31.2973|
> > > > > | GINO  | 30.4572| 30.4674| 30.4697| 30.4697| 30.4693| 30.468 | 30.5234|30.4691| 30.4698|
> > > > >
> > > > > _SSIM:_
> > > > >
> > > > > | Method    |1.0x|1.25x|1.5x |1.75x|2.0x |2.25x| 2.5x |2.75x|3.0x |
> > > > > | -------- | ------- | ------- | ------- | ------- | ------- | ------- | ------- | ------- |------- |
> > > > > | Proposed*  | 0.6808| 0.6812| 0.6826| 0.6847| 0.6872| 0.6898| 0.6962| 0.6948|0.6972|
> > > > > | GINO  | 0.6334| 0.6379| 0.6418| 0.6459| 0.6498| 0.6533| 0.6601| 0.6596|0.6624|
> > > > >
> > > > > Unlike the previous FNO-based experiment, GINO model seems to perform much better. However, our model still outperforms the GINO model performance.
> > > > >
> > > > > We are currently evaluating the GINO model on out-of-distribution super resolution scales. Once we have all the necessary results ready, we are open to incorporate the results of GINO model in our revised manuscript, and also the GINO model's experimental setup (possibly in the Appendix due to space constraint).
> > > > >
> > > > > Please let us know if you have any further questions, comments or suggestions.

---

> > > > > ### Author Response · Authors · 2024-11-24
> > > > > **Revised Manuscript with GINO Results**
> > > > >
> > > > > We would like to thank the reviewer once again for suggesting experiment with Geometry-Informed Neural Operator.
> > > > >
> > > > > The results for in-distribution and out-of-distribution super-resolution scales are added in the revised manuscript. The manuscript has been also updated with details of the experiments with GINO model.
> > > > >
> > > > > Please let us know if you have any further questions, comments or suggestions.

---

> > > > > > ### Comment · Reviewer_Ztio · 2024-11-24
> > > > > > **Thank you for the updates**
> > > > > >
> > > > > > Thank you for the new updates and results on the GINO baseline. I believe the paper's quality has improved, and I have updated my score accordingly.

---

> > > > > > > ### Author Response · Authors · 2024-11-25
> > > > > > > **Thank you for Responding to Our Rebuttal**
> > > > > > >
> > > > > > > We sincerely thank you for taking the time to read our rebuttal and for your thoughtful comments and baseline experimental suggestions. We feel happy that you believe the paper's quality has improved and greatly appreciate your score improvement.
> > > > > > >
> > > > > > > Please feel free to suggest any further questions, comments or suggestions.

---

### Official Review · Reviewer_Pjzg · 2024-11-03

**Soundness:** 2
**Presentation:** 3
**Contribution:** 2
**Rating:** 5
**Confidence:** 4

**Summary:**

This paper presents an implicit neural network-based approach to analyzing wind data, addressing key challenges like limited data resolution (super-resolution downscaling task), high storage demands (data compression), and efficient extrapolation to unmeasured locations (spatial extrapolation).

**Strengths:**

- Meaningful motivation and nice problem settings - addressing several challenges (various resolutions, high-dimension, unobserved areas) in climate data.
- Nice application of Kolmogorov-Arnold Network (KAN) to climate domain.
- Fluent paper representation and enough experiments.

**Weaknesses:**

- Lack of state-of-the-art ML baselines for comparison.
- Several claims are not precise (please see the questions below).
- No available codes.

**Questions:**

- You claim your work is **continuous** super-resolution, how finer is it to say "continuous"?
- I believe there is some related work using diffusion models and generative adversarial networks (GANs) for downscaling tasks. You may want to consider them as ML baselines. The current comparison against ML baselines is relatively weak.
- As you mentioned in the related work, data sources could be text, images, audio, and video. This is why it is called multi-modal learning (generally in the AI domain). In your work, you call the data at different altitudes as different modalities. It is confusing for users, especially for ICLR readers.
- You claim your model enables reconstruction at any scale. Do you use one model or train separate models for different scales?
- PSNR, SSIM, and CR are used as evaluation metrics. It would be better to provide the details on how to compute them? Any equations? Adding them in the appendix also helps if you have a limited-space problem.
- In Figure 4, it is surprising to see the model performance does not drop as the super-resolution scale increases. Could you please provide any possible reasons?
- In section 5.6, you train the ML model on one region and test it on other regions. This is actually an out-of-distribution (OOD) problem. You may want to add a short discussion on it (FYI). And, what is the meaning of the $x$-axis in Figure 6? Why are there some values between 1 and 2 or between 2 and 3?

**Details Of Ethics Concerns:**

NaN

---

> ### Author Response · Authors · 2024-11-20
>
> We thank reviewer Pjzg for the comments. We would like to respond to some of the issues the reviewer mentioned:
>
> _Lack of state-of-the-art ML baselines for comparison:_ Dear reviewer, thank you for your comment on state-of-the-art (SOTA) baselines. Our work tackles three tasks—data compression, modality transfer, and super-resolution—simultaneously. To our knowledge, GEI-LIIF is the only comparable work and is included as a baseline. We have also extended our ablation study (Section 6.6 in the revised manuscript) to evaluate each task individually against SOTA methods specific to those tasks. If you have additional baseline suggestions, we would greatly appreciate them.
>
> _Availability of codes:_ We will release the codes upon acceptance of our paper. However, we are open to provide specific code implementations, e.g. model architecture, before the official release.
>
> _How fine is continuous?_ Thank you for highlighting this concern. By "continuous super-resolution," we refer to our decoder's ability to perform inference at any super-resolution scale or arbitrary coordinate point, unlike traditional models with fixed upsampling ratios. This flexibility justifies the term "continuous super-resolution."
>
> _Multi-altitude vs. multi-modality:_ We define multimodality when data from different modalities are acquired from different sensors. Under this definition, multi-altitude wind data can be attributed to multi-modal data. “Implicit neural representations for simultaneous reduction and continuous reconstruction of multi-altitude climate data”, which we use as the baseline method also follows this definition of multi-modality and attributes multi-altitude data as multimodal data. This definition of multi-modality is added in the revised manuscript. Furthermore, inspired by your insightful comment, we are conducting additional experiments where the input and output consist of distinct climate variables (e.g., temperature and water vapor), representing dual modalities in the earth system. While these experiments are still ongoing, we will update the manuscript with the latest results as soon as they are available.
>
> _Comparison to generative model based downscaling (e.g. diffusion, GAN):_ Traditional downscaling tasks focus on reconstruction to the same dimension as to that of the input. On the contrary, we focus on discrete downscaling and consecutive continuous reconstruction. The problem can be seen as two separate tasks and two separate models can be employed (obviously solved separately). However, we only focus on those approaches that treat both tasks as one single problem. Additionally, in this work, we did not focus on generative model based downscaling. We hope to deal with the same problem of this work with generative models in future.
>
> _One for all scales:_ One model for any arbitrary scale super resolution, offering flexibility over fixed scale super resolution models.
>
> _Description of metrics:_ Added the details in Appendix of the revised manuscript.
>
> _Stable performance across different super resolution scales:_ This is indeed the purpose of designing continuous reconstruction with implicit neural networks, which is expected to capture the functional relationships better in the continuous spatial coordinates compared to traditional regular discrete grid based super-resolution. Implicit neural networks reconstruct the data in a pixel-by-pixel or coordinate-by-coordinate manner in arbitrary spatial coordinates, unlike other super-resolution models with the fixed regular grids. Also, the PSNR metric evaluates the average performance of the reconstruction over all the pixels. That’s why the PSNR evaluations do not change that much over the different super-resolution scales. However, at OOD (out-of-distribution) super-resolution scales, the result might change and it is indeed an area for exploration.
>
> _Out of distribution scales:_ Updated Figure 3 with out-of-distribution super-resolution scales.
>
> _Zero-shot learning/Spatial generalization:_ Added extra experimental results on altitude-based spatial generalization test. Due to space constraints, we have moved the section on spatial generalizability section to Appendix Section D.
>
> _Clarity on figures:_ On all the figures of results, the $x$-axis shows different super resolution scales. In some figures, fractional super-resolution scales may not be shown in the $x$-axis labels to keep the figure cleaner and clearer.
>
> According to the valuable comments and suggestions from all the reviewers, we have updated our manuscript and submitted the revised version. Please let us know if you have any questions or suggestions.

---

> ### Author Response · Authors · 2024-11-24
> **Additional SOTA ML Baselines for Comparison**
>
> We would like to re-address one of the issues raised by the reviewer. According to the suggestion from one of the reviewers on additional baseline experiment, we have added Geometry-Informed Neural Operator based experimental results in the revised manuscript.
>
> _Lack of state-of-the-art ML baselines for comparison:_ Our work tackles three tasks—data compression, modality transfer, and super-resolution—simultaneously. To our knowledge, GEI-LIIF is a comparable work with the highest similarity to our problem and is included as a baseline. We thank the reviewers for suggesting additional baseline experiment with Fourier Neural Operators. We chose Geometry-Informed Neural Operator [5] as the additional baseline experiment due to its suitability for this specific problem statement, and updated the manuscript with results of this additional baseline experiment.
>
> [5] Li, Zongyi, et al. "Geometry-informed neural operator for large-scale 3d pdes." Advances in Neural Information Processing Systems 36 (2024).
>
> Please let us know if you have any further questions, comments or suggestions.

---

> ### Author Response · Authors · 2024-11-27
> **Follow-up on Revised Paper Submission**
>
> Dear Reviewer,
>
> We hope this message finds you well. As the deadline for manuscript revision is approaching to the end, we wanted to follow up to check if you have had a chance to review the revised version of the paper. Please let us know if there are any additional revisions or clarifications required on our part. We would be happy to address them promptly.
>
> Thank you for your time and feedback!
>
> Regards,
>
> Authors

---

### Author Response · Authors · 2024-11-20
**Revision of Manuscript**

We thank all reviewers for their thoughtful and valuable comments. We have revised our manuscript accordingly. The highlights of the major changes are discussed here:

_Revised Problem Statement \& Methodology Section:_ We have updated our **Problem Statement** and **Methodology** section. For better readability, we have added a **Preliminaries** section that discusses some preliminaries on LIIF, Gaussian Adaptive Attention, KAN, Three-dimensional Encoding, and defines why we interpret multi-altitude data as multi-modal data.

_Revised Experimental Setup:_ We have updated the **Experimental Setup** section according to the reviewers' comments for better clarity.

_Additional Experimental Results:_ We have compared our proposed LIIF-KAN with more super-resolution models, including:
 - HiNOTE ([1] ICML 2024)
 - SRNO ([2] CVPR 2023)
 - DIINN ([3] WCAV 2023)
 - LTE-KAN ([4] CVPR 2022)

along with the previous baselines. As our problem statement specifically requires continuous reconstruction, we only focus on existing continuous super-resolution models as baselines.

[1] Luo, X., Qian, X. and Yoon, B.J., Hierarchical Neural Operator Transformer with Learnable Frequency-aware Loss Prior for Arbitrary-scale Super-resolution. In Forty-first International Conference on Machine Learning.

[2] Wei, M. and Zhang, X., 2023. Super-resolution neural operator. In Proceedings of the IEEE/CVF Conference on Computer Vision and Pattern Recognition (pp. 18247-18256).

[3] Nguyen, Q.H. and Beksi, W.J., 2023. Single image super-resolution via a dual interactive implicit neural network. In Proceedings of the IEEE/CVF Winter Conference on Applications of Computer Vision (pp. 4936-4945).

[4] Lee, J. and Jin, K.H., 2022. Local texture estimator for implicit representation function. In Proceedings of the IEEE/CVF conference on computer vision and pattern recognition (pp. 1929-1938).

We have also added extra experimental results on altitude-based spatial generalization test. Due to space constraints, we have moved the section on spatial generalizability section to Appendix Section D.

We have also updated Figure 3 with out-of-distribution super-resolution scales.

_Revised Baseline Experimental Result:_ We thank the reviewers for suggesting additional baseline experiment with Fourier Neural Operators. We chose Geometry-Informed Neural Operator [5] as the additional baseline experiment due to its suitability for this specific problem statement, and updated the manuscript with results of this additional baseline experiment.

[5] Li, Zongyi, et al. "Geometry-informed neural operator for large-scale 3d pdes." Advances in Neural Information Processing Systems 36 (2024).

_Neat figures:_ We have updated all the figures in the revised manuscript with hopefully improved figures for more visibility and readability.

_Notation update:_ We have added a list of notations in the Appendix Section B to make the notations easy to follow.

_Description of metrics:_ We have added the details of the evaluation metrics in Appendix section C of the revised manuscript.

If you have additional thoughts in mind, we would greatly appreciate your suggestions.

---

### Author Response · Authors · 2024-11-25

Dear Reviewers,

As the author-reviewer discussion period is approaching the deadline soon, we kindly request you to review our responses to your comments, concerns and suggestions. If you have further questions or comments, we will do our best to address them before the discussion period ends. If our responses have resolved your concerns, we would greatly appreciate it if you could update your evaluation of our work accordingly.

Thank you once again for your valuable time and thoughtful feedback.

Regards,

Authors

---

### Meta-Review · Area_Chair_G5YZ · 2024-12-19

**Metareview:**

The paper presents a new architecture for analyzing wind data. The model is able to superresolve the wind fields to any resolution (through continuous representation), functions as a data compressor (through downsampling layers), and allows for modality transfer to different altitude levels. The paper presents experiments demonstrating the model's strengths in the above three tasks, ablations motivating the components of the architecture.

Strengths: new architecture design that is well motivated for the problem, experiments and ablations demonstrating the effectiveness of the architecture.

Weaknesses: the contributions seem incremental  - while the compression results are positive, the inclusion of other baselines like GINO show that other architectures can also show good performance on the other tasks. Another weakness is the dense presentation of material in the paper.

An analysis on the impact of the performance improvement over other models (say GINO) will be useful. Showing careful visual predictions that highlight the downscaling advantages can also be useful. The authors could additionally consider metrics like power spectra plots that show what wavenumbers are adequately reconstructed by their model and what other models miss. Superresolution in physical tasks may require a deeper analysis on what is being reconstructed rather than averaged metrics.

**Additional Comments On Reviewer Discussion:**

The reviewers raised multiple concerns about other ML baselines. I believe the authors have addressed them suitably. Generative models could also be considered as baselines since they are quite common in downscaling tasks in the climate+AI domain.
There were several other technical questions from the reviewers as well as suggestions on improving the presentation - the authors have made good effort in answering the questions as well as revising their manuscript with more baselines.
However, due to the weaknesses highlighted above, the reviewers continue to keep their score and I am in general agreement.

---

### Decision · Program_Chairs · 2025-01-22

Reject